# Analytical constraints on layered gas trapping and smoothing of atmospheric variability in ice under low accumulation conditions

Kévin Fourteau[1], Xavier Faïn[1], Patricia Martinerie[1], Amaëlle Landais[2], Alexey A. Ekaykin[3], Vladimir Ya. Lipenkov[3], and Jérôme Chappellaz[1]

[1]Univ. Grenoble Alpes, CNRS, IRD, Grenoble INP, IGE, F-38000 Grenoble, France
[2]Laboratoire des Sciences du Climat et de l'Environnement, UMR8212, CEA-CNRS-UVSQ-UPS/IPSL, Gif-sur-Yvette, France
[3]Climate and Environmental Research Laboratory, Arctic and Antarctic Research Institute, St. Petersburg, 199397, Russia

*Correspondence to:* kevin.fourteau@univ-grenoble-alpes.fr or patricia.martinerie@univ-grenoble-alpes.fr

**Abstract.** We investigate for the first time through continuous measurements the loss and alteration of past atmospheric information from air trapping mechanisms under low accumulation conditions. Methane concentration changes were measured over the Dansgaard-Oeschger event 17 (~~D0-17~~DO-17, $\sim 60,000\,\mathrm{yr BP}$) in the Antarctic Vostok 4G-2 ice core. Measurements were performed using continuous-flow analysis combined with laser spectroscopy. The results highlight many anomalous layers at the centimeter scale, unevenly distributed along the ice core. The anomalous methane mixing ratios differ from those in the immediate surrounding layers by up to $50\,\mathrm{ppbv}$. This phenomenon can be theoretically reproduced by a simple layered trapping model, creating very localized gas age scale inversions. We propose a method for cleaning the record of anomalous values which aims at minimizing the bias in the overall signal. Once the layered-trapping induced anomalies are removed from the record, the DO-17 appears to be smoother than its equivalent record from the high accumulation WAIS Divide ice core. This is expected due to the slower sinking and densification speeds of firn layers at lower accumulation. However and surprisingly, the degree of smoothing appears similar between modern and DO-17 conditions at Vostok. This suggests that glacial records of trace gases from low accumulation sites in the East Antarctic plateau can provide a better time resolution of past atmospheric composition changes than usually expected. We also developed a numerical method to extract the gas age distributions in ice layers ~~that~~ based on the comparison with a weakly smoothed record. It can be applied even for sites without firn-air measurements. It is particularly adapted for the conditions of the East Antarctic plateau, as it helps to characterize smoothing for a large range of very low temperature and accumulation conditions.

## 1 Introduction

In a context of climate change, the study of paleoclimate is an important tool for understanding the interactions between climate and atmospheric conditions (Masson-Delmotte et al., 2013). Ice cores have been used to retrieve climatic and atmospheric conditions back to $800,000\,\mathrm{yr}$ before present (BP) (Jouzel et al., 2007; Loulergue et al., 2008; Lüthi et al., 2008). Notably, ancient

atmospheric gases get enclosed within bubbles in the ice material and allow ~~reconstructing~~ us to reconstruct the past history of atmospheric composition (Stauffer et al., 1985). The trapping of air in ice is due to the transformation of firn (porous compacted snow) into airtight ice at depths ranging from $\sim 50$ to $\sim 120\,\mathrm{m}$ depending on temperature and accumulation conditions. It is characterized by an increase in bulk density and a decrease in porosity with depth along the firn column. It is only at the bottom of the firn column that the porosity of the medium gets closed and traps the interstitial air. From a gas point of view the firn is traditionally divided in three main parts from surface to bottom: the convective zone, the diffusive zone and the trapping zone (e.g. Schwander, 1989; Buizert et al., 2012). The convective zone is characterized by the mixing of air in the firn porosity with atmospheric air through wind action (Colbeck, 1989). In the diffusive zone the dominant gas transport process is molecular diffusion with additional contribution from gravitational settling. Finally, the trapping zone corresponds to the enclosure of air into bubbles through the closure of the porosity. The process of densification and pore closure can last for thousands of years at the most arid sites in Antarctica.

Air trapping affects the recording of atmospheric ~~composition events~~ variability in ice cores. One known effect of gas enclosure mechanism is the ~~dampening~~ damping of fast variations in the atmosphere, also called smoothing (Spahni et al., 2003; Joos and Spahni, 2008; Köhler et al., 2011; Ahn et al., 2014). This smoothing arises from two reasons: (i) the gas diffusion in the firn mixes air from different dates, and thus a bubble does not enclose gases with a single age but rather an age range (Schwander et al., 1993; Rommelaere et al., 1997; Trudinger et al., 1997; Witrant et al., 2012); (ii) in a given horizontal layer, bubble enclosure takes place over a range of time rather than a precise instant. These two phenomena combined mean that at a given depth, the air enclosed is represented by a gas age distribution, and not by a single age (Schwander et al., 1993; Rommelaere et al., 1997). Gas enclosure mechanisms thus act as a low-pass filter, attenuating signals whose periods are too short compared to the span of the distribution. Spahni et al. (2003) reported the only existing observations of the smoothing effect under low accumulation conditions. They concluded that the abrupt methane variation during the cold event of $8.2\,\mathrm{kyrBP}$ recorded in the EPICA Dome C ice core, compared with its counterpart from the Greenland GRIP ice core had experienced an attenuation of $34\%$ to $59\%$. Sites with low accumulation tend to have broader age distributions leading to a stronger ~~dampening~~ damping effect (Spahni et al., 2003; Joos and Spahni, 2008; Köhler et al., 2011; Ahn et al., 2014). A heuristic explanation is that the span of the age distribution is directly related to the densification speed of a firn layer, which is slow at the low temperature and arid sites of the Antarctic plateau. Moreover, for the most arid sites the impact of diffusive mixing is negligible compared to progressive trapping, and the smoothing is hence mainly driven by the speed of porosity closure.

Even if the bulk behavior in firn is the increase of density and decrease of open porosity with depth, local physical heterogeneities affect firn densification and gas trapping (Stauffer et al., 1985; Martinerie et al., 1992; Hörhold et al., 2011; Fujita et al., 2016). Working on ice cores and firn from high accumulation sites Etheridge et al. (1992), Mitchell et al. (2015) and Rhodes et al. (2016) have discussed the influence of ~~short~~ centimeter scale physical variability in firn on recorded gas concen-

trations. They argue that physical heterogeneities can lead to variations in closure depth for juxtaposed ice layers. For instance a given layer could reach bubble enclosure at shallower depth and earlier (respectively deeper and later) than the surrounding layers in the firn, thus trapping relatively older gases (respectively younger gases). In periods of ~~fast~~ atmospheric variations in trace gases composition occurring at a similar time scale as the trapping process, this mechanism can lead to gas concentration anomalies along depth in an ice core and has been called layered bubble trapping. Based on observations in high accumulation

Greenland ice cores, and modeling for the WAIS Divide ice core, Rhodes et al. (2016) report that such artifacts can reach $40\,\mathrm{ppbv}$ in the methane ($CH_4$) record during the industrial time~~in high accumulation ice cores from Greenland and the WAIS Divide Antarctic ice core~~. In addition, the amplitude of the artifacts increases with lower accumulation rates.

Here we investigate for the first time the existence and impacts of heterogeneous trapping and smoothing in very low

accumulation conditions using continuous measurements of trace gases. High resolution methane concentration (combined with carbon monoxide) measurements were performed along a section of the Vostok 4G-2 ice core, drilled in the Antarctic plateau. The section studied corresponds to the Dansgaard-Oeschger event number 17 (DO-17, $\sim 60,000\,\mathrm{yrBP}$), a climatic event associated with particularly fast and large atmospheric methane variations (Brook et al., 1996; Chappellaz et al., 2013; Rhodes et al., 2015). This makes this event especially adapted for the quantification of both gas record smoothing and layered

trapping. To interpret our data we compare them with the much less smoothed methane record measured in the WAIS Divide ice core ~~(Rhodes et al., 2015)~~ (WDC, Rhodes et al., 2015) , where the accumulation rate is an order of magnitude larger than at Vostok.

## 2 Ice core samples and analytical methods

### 2.1 Vostok ice samples

The ice core analyzed in this study is the 4G-2 core drilled at Vostok, East Antarctica in the 1980s (Vasiliev et al., 2007). Measured depths range from $895$ to $931\,\mathrm{m}$, with a cumulative length of $27.5\,\mathrm{m}$ due to several missing portions in the archived ice at Vostok station. The ice core sections analyzed have been stored at Vostok Station since the drilling, and were transported

to Institut des Geosiences de l'Environnement (IGE, Grenoble, France; formerly LGGE) 3 months before analyses. Although stored at Vostok at temperatures of $\sim -50\,°C$, the samples showed clathrate relaxation cavities. The gas age over this depth interval spans over a $3,000\,yr$ interval centered on $59,400 \pm 1,700\,yrBP$ (Bazin et al., 2013; Veres et al., 2013). ~~It was selected to include the Dansgaard-Oeschger event 17, showing a rapid and large increase in atmospheric methane concentration of about~~

~~$150\,ppbv$ within $500\,yr$ (Brook et al., 1996; Chappellaz et al., 2013; Rhodes et al., 2015) .~~ The estimated snow accumulation rate at the Vostok core site for this period is $1.3 \pm 0.1 \mathrm{cm\,ice\,yr^{-1}}$ (Bazin et al., 2013; Veres et al., 2013). Even though DO events are associated with large warmings in the northern hemisphere, isotopic records indicate that DO-17 temperatures on the Antarctic plateau remain at least 5°C below modern temperatures (Figure 2 in Jouzel et al., 2007).

## 2.2 Continuous methane measurements

The Vostok ~~4G2~~ 4G-2 ice core sections were analyzed at high resolution for methane concentration (as well as carbon monoxide as a by-product) at IGE over a 5-days period and using a continuous ice core melting system with online gas measurements (CFA, continuous flow analysis). Detailed descriptions of this method have been reported before (Stowasser et al., 2012; Chappellaz et al., 2013; Rhodes et al., 2013). Briefly, ice core sticks of $34$ by $34\,mm$ were melted at IGE at a mean rate of ~~$3.8\,cm.min^{-1}$~~ $3.8\,\mathrm{cm\,min^{-1}}$ using a melt head as described by Bigler et al. (2011), and the water and gas bubble mixture

was pumped toward a low volume T-shaped glass debubbler. All the gas bubbles and approximately $15\%$ of the water flow were transferred from the debubbler to a gas extraction unit maintained at $30\,°C$. The gas was extracted by applying a pressure gradient across a gas-permeable membrane (optimized IDEX in-line degasser, internal volume $1\,mL$). The gas pressure recorded downstream of the IDEX degasser was typically $500 - 600\,mbar$ and was sufficiently low to extract all visible air bubbles from the sample mixture. A home-made Nafion dryer with a ~~$30\,mL.min^{-1}$~~ $30\,\mathrm{mL\,min^{-1}}$ purge flow of ultra-pure

nitrogen (Air Liquide $99.9995\%$ purity) dried the humid gas sample before entry into the laser spectrometer. Online gas measurements of methane were conducted with a SARA laser spectrometer developed at Laboratoire Interdisciplinaire de Physique (Grenoble, France) based on Optical Feedback-Cavity Enhanced Absorption Spectroscopy (OF-CEAS, Morville et al., 2005; Romanini et al., 2006). Such a laser spectrometer has been used before for continuous flow gas analyses (e.g., Chappellaz et al., 2013; Rhodes et al., 2013, 2015, 2016; Faïn et al., 2014); however, the IGE CFA system was specifically optimized to

reduce experimental smoothing by limiting all possible dead and mixing volumes along the sample line. For this study the rate of OF-CEAS spectrum acquisition was $6\,Hz$. The $12\,cm^3$ optical cavity of the spectrometer was maintained at $30\,mbar$ internal pressure, which corresponds to an equivalent cavity volume of only $0.36\,cm^3$ at STP and allows for a fast transit time of the gaseous sample in the cavity. Consequently, the SARA instrument introduces a significantly lower smoothing than the CFA

setup. The SARA spectrometer was carefully calibrated onto NOAA2004 scale (Dlugokencky et al., 2005) before the CFA analyses using three synthetic air standards with known methane concentrations (Scott Marrin Inc., Table S1, Supplementary Information (SI)). $CH_4$ concentrations measured during the calibration agreed with NOAA measurements within $0.1\%$ over a 360-1790 ppbv range. A linear calibration law was derived and applied to all $CH_4$ data (Figure S1, SI).

Allan variance tests (Allan, 1966; Rhodes et al., 2013) were conducted using mixtures of degassed deionized water and synthetic air standard to evaluate both the stability and the precision of the measurements. The best Allan variance was obtained on an integration time larger than 1000 s, illustrating the very good stability of the CFA system. However, in order to optimize the depth resolution of our measurements, we used an integration time of 1 s for which a precision of 2.4 ppbv ($1\sigma$) was observed. This corresponds to a peak-to-peak $CH_4$ variability of $\sim 10$ ppbv. Hereafter, this variability will be referred as analytical noise.

The mixing of gases and melt water during the sample transfer from the melt head to the laser spectrometer induces a CFA-experimental smoothing of the signal. The extent of the CFA based ~~dampening~~ damping was determined by performing a step test (left panel of Figure S2, Supplement), i.e. a switch between two synthetic mixtures of degassed DI water and synthetic air standards of different methane concentrations, following the method of Stowasser et al. (2012). It shows that the CFA system can resolve signals down to the centimeter scale. We were also able to extract the impulse response of the system, that will

be used in Section 4.3 to emulate CFA smoothing. A more detailed discussion of the frequency response of the system can be found in Section S2 of the Supplement. Breaks along the core regularly let ambient air enter the system, resulting in strong positive spikes in methane concentration. In order to remove these contamination artifacts, exact times corresponding to a break running through the melt head were recorded during the measurements and later used to identify and clean the data from contamination.

**2.3   Nitrogen isotopes**

The ratio of stable nitrogen isotopes, $^{15}N/^{14}N$, was measured at Laboratoire des Sciences du Climat et de l'Environnement (LSCE), France. Briefly, a melting technique followed by gas condensation in successive cold traps was used to extract the air from the ice, and the air samples were then transferred to a dual inlet mass spectrometer (Delta V plus, Thermo Scientific). The analytical method and corrections applied to the results are described in Landais et al. (2004), and references therein. The

results are expressed as deviations from the nitrogen isotopic ratio in dry atmospheric air ($\delta^{15}N$). Discrete samples every 50 cm and duplicates were analyzed when possible. A total of 96 data points, including 39 duplicates were obtained. The pooled standard deviation over duplicate samples is $0.011\%_o$.

# 3 Experimental results

## 3.1 Methane record

The methane record spanning over the DO-17 event extracted from the Vostok ~~4G2~~ 4G-2 ice core is presented in blue in Figure 1. Two corrections were applied to these data: (i) data screening and removal of kerosene contaminations, and (ii) full dataset calibration to account for the preferential dissolution of methane during the melting process.

Kerosene, used as drilling fluid for the Vostok 4G-2 ice core extraction, was detected in some of the meltwater from our continuous flow analysis. This contamination induces surface iridescent colors and a strong characteristic smell, and was detected not only on the meltwater from the outer part of our ice sticks but also in some of the meltwater from the center of the ice samples. However, the continuous flow of the meltwater does not allow us to clearly identify the contaminated ice core sections. Carbon monoxide (CO) was measured simultaneously with methane by our laser spectrometer (Faïn et al., 2014). We attributed simultaneous anomalies in $CH_4$ (increase of about $20\,\mathrm{ppbv}$ or more) and CO (increase of about $100\,\mathrm{ppbv}$ or more) mixing ratios to kerosene contaminations, and suppressed the corresponding data by visual inspection of the dataset. An example of such a kerosene contamination is visible Figure S3 (SI). Chappellaz et al. (1990) indicate that methane contaminations lower than $40\,\mathrm{ppbv}$ were observed by discrete measurements in the brittle zone of the Vostok 3G core, consistently with our observation in 4G-2. The impact of kerosene contamination on CO in ice cores has not been quantified so far. ~~Overall, data along 2.1 m of ice sections were removeddue to kerosene contamination~~Adding the length of all kerosene affected ice core sections, a total of $2.1\,\mathrm{m}$ of data was removed. The calibration of methane mixing ratio for preferential solubility (Rhodes et al., 2013) was achieved by matching our continuous methane measurements with the already calibrated ~~WAIS~~ WDC methane data set (Rhodes et al., 2015), as described in Section S1.2 of Supplementary Information. The resulting methane record has a high resolution, but presents numerous discontinuities due to missing ice, ambient air infiltrations, and kerosene contaminations. The signal displays two distinct scales of variability.

Atmospheric history relevant variability: The general shape of the signal can be divided in two parts, a stable zone extending from 931 to $915\,\mathrm{m}$ depth, then two consecutive methane variations of approximately $100\,\mathrm{ppbv}$ each, extending respectively from 915 to $907\,\mathrm{m}$ and from 907 to $895\,\mathrm{m}$. They respectively correspond to the plateau preceding the DO-17 event, and the DO-17 event itself.

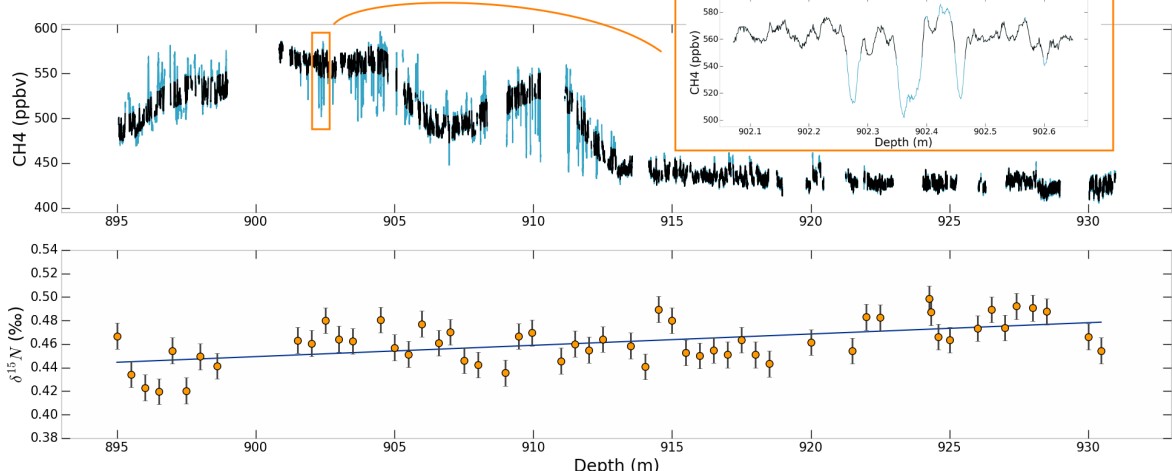

**Figure 1.** Top: Methane concentration along the Vostok 4G-2 ice core. In blue: data cleaned from ambient air and kerosene contamination, and calibrated. In black: data cleaned from layered trapping. Top right corner: zoom over the section from $902.0$ to $902.7\,\mathrm{m}$.
Bottom: $\delta^{15}N$ of $N_2$ as a function of depth in the Vostok 4G-2 ice core. Orange dots: isotopic measurements. The vertical error bars correspond to the pooled standard deviation. In blue: linear regression.

Centimeter scale variability: The signal also displays centimeter scale methane variations. A portion of these variations is explained by the $10\,\mathrm{ppbv}$ analytical noise of the CFA system. However, in the upper part of the core (above $915\,\mathrm{m}$) the signal also exhibits abrupt variations with amplitudes up to $50\,\mathrm{ppbv}$ ~~with~~ and widths of about $2\,\mathrm{cm}$. Most of those spikes are negatively orientated and therefore laboratory air or kerosene contamination can be ruled out. It should be noted that the width

of the spikes are in the attenuation range of the CFA system, meaning that the true signal in the core has a somewhat larger amplitude than the measured signal. Moreover, the spikes exhibit a specific distribution with depth. For instance no spike is observed in the lower part of the ice core where the methane concentration is essentially flat, and only negative spikes appear in between $900$ and $905\,\mathrm{m}$ depth as seen in the zoomed part Figure 1.

**3.2    Revised age scale using Nitrogen isotopes**

The current reference chronology for the Vostok ice core is the Antarctic Ice Core Chronology 2012 (AICC2012; Bazin et al., 2013; Veres et al., 2013). However, only two gas stratigraphic links between Vostok and other cores are available for the DO-17 period in AICC2012, leading to relatively large uncertainties in the Vostok gas age scale over this period. The $\delta^{15}N$ of $N_2$ profile over DO-17 event in the Vostok core is shown Figure 1. We fitted the experimental values with a linear regres-

sion (slope of ~~$9.63 \times 10^{-4}\,\%o.\mathrm{m}^{-1}$~~ $9.63 \times 10^{-4}\,\%o\,\mathrm{m}^{-1}$ and intercept of $-0.417\%o$). Considering the diffusive zone of the

firn to be stratified according to a barometric equilibrium (Craig et al., 1988; Orsi et al., 2014), its height can be expressed as $H = (RT/g\Delta M)\ln(1 + \delta^{15}N)$, where $R$ is the ideal gas constant, $T$ the temperature, $g$ the gravitational acceleration, and $\Delta M$ the difference in molar mass between $^{14}N$ and $^{15}N$. With a firn temperature of $217\,K$ (Petit et al., 1999), the mean $\delta^{15}N$ value of $0.46\%_o$ translates into a diffusive column height of $85\,m$, and a LIDIE (lock-in depth in ice equivalent) of $59\,m$ (using a mean firn relative density of $0.7$). This value lies in the lower range of the AICC2012 LIDIE estimations for this depth range in the Vostok ice core: $58$ to $70\,m$ (Bazin et al., 2013; Veres et al., 2013).

The age difference between the ice and the enclosed gases ($\Delta$Age) can be estimated using the height of the firn with: $\Delta$Age $=(H + H_{conv})D/accu$, where $H$ and $H_{conv}$ are respectively the heights of the diffusive and convective zones, $D$ is the average density of the firn column and $accu$ the accumulation rate. Present-day observations report a convective zone spanning down to $13\,m$ at Vostok (Bender et al., 1994). We used this value as an estimate for the convective zone depth during the DO-17. In Figure 2, $\Delta$Age values inferred from our $\delta^{15}N$ record, using $D = 0.7$ and an accumulation rate of $1.3\,cm\,ice\,yr^{-1}$, are compared with the values from AICC2012 (Bazin et al., 2013; Veres et al., 2013). The AICC2012 $\Delta$Age values display a variability of several centuries as shown by the dashed black line in Figure 2. These variations are sufficient to induce significant distortions in the duration of methane events. These distortions affect the comparison between our measurements and the WAIS WDC record from Rhodes et al. (2015), as seen in Figure S11 of the supplement. Furthermore, the amplitude of the $\Delta$Age variations is similar to the uncertainty on gas age ($1479$ to $1841$ years). The studied period is fairly stable in terms of temperature and accumulation at Vostok (Petit et al., 1999; Bazin et al., 2013; Veres et al., 2013), thus the $\Delta$Age changes in the AICC2012 chronology are likely to result from artifacts of the optimization method rather than to correspond to actual variations. We hence revised the AICC2012 gas age scale, by deriving a new smooth gas age using AICC2012 ice age scale and our $\Delta$Age values inferred from the linear interpolation of $\delta^{15}N$ data (Figure 2). This new smooth chronology enables to visually identify the different sub-parts of the DO-17 event between the Vostok and WAIS methane record WDC methane records. It is important to note that this gas age chronology will be again improved by matching the Vostok and WAIS WDC methane records (see Section 5.2). The corresponding $\Delta$Age of this final chronology is displayed as the green line in Figure 2.

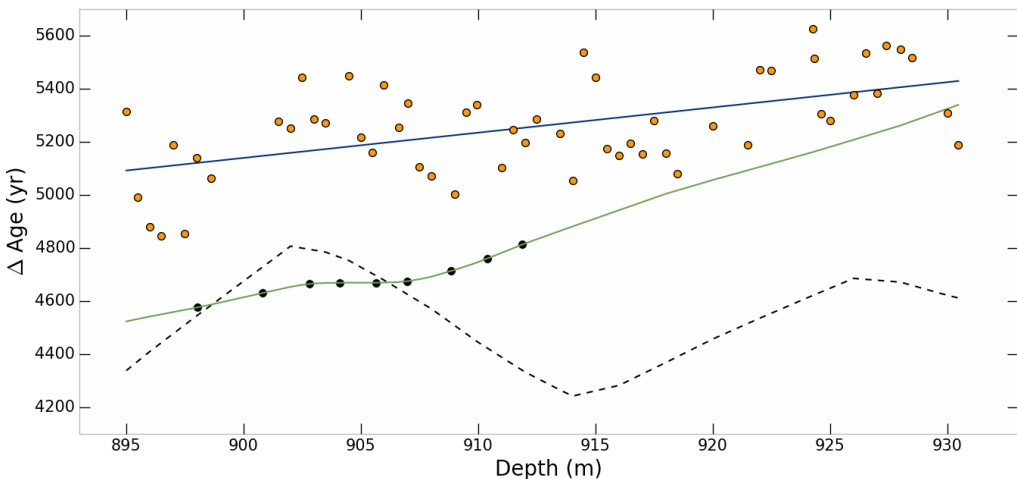

**Figure 2.** ~~Δ*Age*~~ ΔAge along the Vostok record. Orange dots: ΔAge directly estimated from $\delta^{15}$N measurements. In blue: ~~Δ*Age*~~ ΔAge derived from the linear regression on isotopic measurements. Black dashed line: ~~Δ*Age*~~ ΔAge as given by AICC2012. In green: ΔAge after matching with the ~~WAIS~~ WDC CH$_4$ record. Black dots: tie points (minima, maxima and mid-slope points) used to match the ~~WAIS~~ WDC record (see 5.2).

## 4   Layered bubble trapping

### 4.1   Conceptual considerations of the layered trapping mechanism

Due to heterogeneities in firn density and porosity, an ice layer may undergo early gas trapping (Etheridge et al., 1992; Rhodes et al., 2013; Mitchell et al., 2015; Rhodes et al., 2016)~~. The closure of such a layer is likely progressive~~, with the pore closure
5   process starting in advance. Thus during gas trapping, the corresponding layer is at a more advanced state of closure than the surrounding bulk layers. Similarly, some layers may undergo a late closure. If gases can circulate through the open porosity surrounding the anomalous layers, the early closed layers will contain abnormally ancient gas with respect to the surrounding layers. On the other hand layers closed late will contain abnormally recent gas. This leads to very local inversions of the gas age scale along depth. As explained in Rhodes et al. (2016), such a mechanism affects trace ~~gases record~~ gas records only during
10   periods of ~~significant atmospheric variations~~ variations in concentration of atmospheric gases. Then, abnormal layers contain air significantly different in composition from surrounding layers and appear as spikes in the record. On the other hand during periods without atmospheric variations, the abnormal layers do not contain air significantly different in composition from their surroundings, and the gas record is not affected.

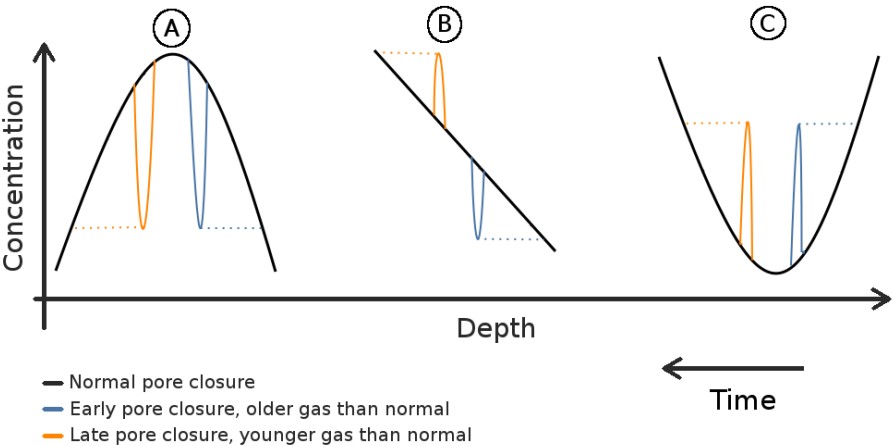

**Figure 3.** Expected orientation of layered trapping artifacts depending on the characteristics of atmospheric variations. Black curves correspond to a normal chronological trapping, blue to early pore closure and orange to late pore closure. Cases A, B and C respectively represent local maximum, monotonous trend and local minimum situations.

The orientation of layered trapping spikes depends on the type of atmospheric variations, as illustrated in Figure 3. For instance in a period of local maximum in methane concentrations, both early and late closures tend to enclose air with lower mixing ratios, as displayed in case A in Figure 3. Similarly, in periods of methane minima, abnormal layers tend to enclose air with larger mixing ratios, as displayed in case C in Figure 3. In the case of monotonous ~~variations~~increase/decrease, early
and late closures lead to artifacts with opposite signs, represented as case B in Figure 3. It should be noted that early and late closure are not expected to affect the record with the same importance. Indeed, ~~during gas trapping in a late closure layer, the~~ ~~surrounding ice will be at least partially impermeable, thus limiting air renewal in this layer. We thus expect early trapping to~~ ~~be dominant compared to late trapping~~a late pore closure means that the surrounding firn is sealed and prevents long distance gas transport. The latest closure layers will not be able to trap young air if gas transport is impossible in the surrounding firn
layers, resulting in less important artifacts.

## 4.2   Observed layered trapping in the Vostok 4G-2 ice core

The positive and negative spikes observed in the Vostok 4G-2 methane record, introduced in Section 3.1, are consistent with the expected impacts of layered trapping. First the absence of spikes in the lower part of the record, below $915\,\mathrm{m}$, is consistent
with the absence of an overall methane trend over the corresponding period. Moreover, in periods of methane local maxima, around 903 and $910\,\mathrm{m}$ ~~depths~~depth, most of the spikes are negatively oriented as expected with the conceptual mechanism of

layered bubble trapping (cf. case A in Figure 3).

Thin sections of ice, covering the depth range between $902.0$ and $902.42\,\mathrm{m}$ (zoomed range on Figure 1) have been analyzed, to investigate if structural anomalies were associated with anomalous trapping. The method is described in detail in Section S4 of the Supplement. We were not able to observe any link between the grain sizes and abnormal layers in the methane record. Nonetheless, structural anomalies may have existed at time of pore closure before disappearing with $\sim 60,000$ years of grain evolution. Other explanations of the methane anomalies than layered trapping were considered as well. Looking for a correlation between ice quality and methane anomalies was also a motivation for the above thin section analysis. Although the samples showed small clathrate relaxation cavities, the CFA sticks did not reveal visual ~~anomalies~~signs of stratification possibly associated with abnormal layers. Examples of a CFA stick picture and thin section results are provided in the Supplement. The ice samples were not large enough to allow for CFA duplicate analysis but the sticks were not melted in a regular depth order so that instabilities in the measurement system could be more easily detected. As contamination cannot explain negative methane concentration anomalies, we could not find a convincing alternate explanation to layered bubble trapping for our results. Contrary to Rhodes et al. (2016) , a spectral analysis of the detrended noise (CFA data points minus spline values) did not show any spike around annual, decadal or any other time scale in our data.

### 4.3 Simple model of layered trapping

A major difficulty for understanding the gas trapping in ice is to relate structural properties measured on small samples to the three dimensional behavior of the whole firn. For example pore closure anomalies have been associated to tortuosity anomalies, with more tortuous layers closing earlier (Gregory et al., 2014), or to density anomalies, with denser layers closing earlier (Etheridge et al., 1992; Mitchell et al., 2015; Rhodes et al., 2016). In this section we used the ~~later~~latter hypothesis, supported by observed relationships between local density and closed porosity (e.g. Stauffer et al., 1985; Mitchell et al., 2015), to test if density driven anomalies could result in artifacts as observed in the Vostok methane record.

In our simple model, the ice core is discretized in layers of $2\,\mathrm{cm}$ width. Abnormal layers are stochastically distributed along the ice core. Based on the characteristics of our Vostok methane signal, we use a density of 10 abnormal layers per meter. They are given a random density anomaly ($\Delta\rho$, normally distributed) representing the density variability at the bottom of the firn. Hörhold et al. (2011) ~~report values of density variability in the closure zone at several sites and derive linear relationships with the site accumulation and temperature. Extrapolating these results to the Vostok trapping zone under glacial conditions, we~~

~~evaluate the standard deviation of density variability to range between 3 and 7 kg.m$^{-3}$~~propose linear regressions of the close-off density variability as a function of accumulation and temperature, based on various sites. Their lowest accumulation site is Dome C, with an accumulation of $2.5\,\mathrm{cm\,ice\,yr^{-1}}$ and a density variability ($\Delta\rho$) of $4.6\,\mathrm{kg\,m^{-3}}$. Applied to Vostok DO-17 conditions, the accumulation based extrapolation leads to a variability of $7\,\mathrm{kg\,m^{-3}}$ and the temperature based extrapolation leads to a variability of $2.7\,\mathrm{kg\,m^{-3}}$. This defines our extreme values (7 and $3\,\mathrm{kg\,m^{-3}}$), and we chose the middle number of $5\,\mathrm{kg\,m^{-3}}$ as the best guess value. Hence, in the model, the abnormal layers are given a firn density anomaly distributed according to a zero-centered Gaussian distribution of standard deviation of ~~5 kg.m$^{-3}$~~$5\,\mathrm{kg\,m^{-3}}$. In order to convert density anomalies into a closure depth ~~shift~~anomaly (the difference in pore closure depth between an abnormal layer and an adjacent layer following the bulk behavior), we assume that all layers have similar densification rates ($d\rho/dz$). Using the data based density profiles at ~~the coldest~~ Dome C, Vostok, and Dome A sites in Bréant et al. (2016), $d\rho/dz$ in deep firn is estimated to be in the range $1.7$ to ~~2.5 kg.m$^{-4}$~~$2.5\,\mathrm{kg\,m^{-4}}$. Thus, the gradient is set to be ~~2 kg.m$^{-4}$~~$2\,\mathrm{kg\,m^{-4}}$. Specifically, a layer closing in advance (or late) closes higher (or lower) in the firn. ~~Combining~~Dividing the above typical density anomaly ~~and depth gradient~~ ($\Delta\rho$) by the depth gradient ($d\rho/dz$), the characteristic depth ~~shift~~anomaly in deep firn of anomalous layers is about 2.5 meters. Using the estimated accumulation rate of $1.3\,\mathrm{cm\,ice\,yr^{-1}}$ for this period, it translates into an age ~~shift~~anomaly (the gas age difference between an abnormal layer and an adjacent layer following the bulk behavior) of about 207 years. To take into account the asymmetry between early and late closure, we reduce the latter's standard deviation of age anomalies to $52\,\mathrm{yr}$ (25% of the early closure anomaly). This is meant to reflect the limitation of air renewal in late closure layers, when the surrounding porosity is already closed and prevents air transport. The value of 25% has been chosen to limit late trapping artifacts in a visually consistent manner with the observations. The methane mixing ratio at a given depth is computed using an atmospheric trend history and a gas age distribution (GAD) of trapped gases (Rommelaere et al., 1997). The atmospheric methane scenario used is the high resolution methane record from the WAIS Divide ice core (Rhodes et al., 2015)~~, with gas ages converted on the AICC2012 scale (Buizert et al., 2015)~~. The WDC gas age chronology (WD2014) was scaled to the GICC05 chronology (with present defined as 1950) dividing by a factor of 1.0063 as in Buizert et al. (2015) . For the rest of the article we used this scaled WD2014 chronology to express WDC gas ages. All layers are supposed to have the same GAD, simply centered on different ages. The GAD used here is the one derived in Section 5.2, specifically for the Vostok ice core during the DO-17 event. A sensitivity test using a very different GAD is described in the next paragraph. Finally, in order to reproduce the gas mixing in the CFA system discussed Section 2.2, the modeled concentrations have been smoothed by convolving the signal with an estimated impulse response of the CFA system (Figure S2, SI). The smoothing characteristics of our measurement system were determined experimentally as in Stowasser et al. (2012). The CFA smoothing induces a ~~dampening~~damping of about 18% of

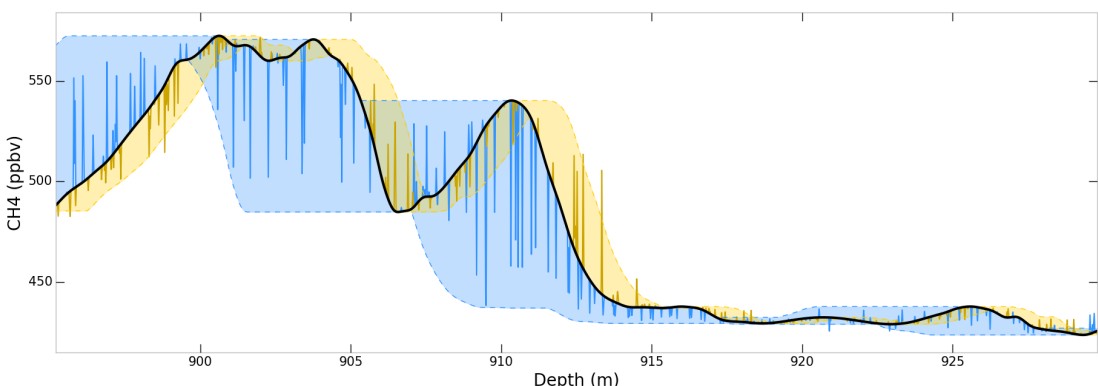

**Figure 4.** Modelled layered trapping artifacts. The black curve represents the results of smooth trapping. Spikes correspond to a single stochastic realization of the layered trapping with CFA smoothing. Blue color stands for early closure and yellow for late closure. Blue shaded areas correspond to the range of concentration anomalies for early closure anomalies up to two standard deviations (depth ~~shift~~ anomaly of 5 m corresponding to an age anomaly of 415 yr). Yellow shaded areas correspond to late closure anomalies with 25% of the early closure extent (depth ~~shift~~ anomaly of 1.25 m corresponding to an age anomaly of 104 yr).

the modeled artifacts.

    The modeled artifacts (Figure 4) globally reproduce well the depth distribution and amplitude of the methane anomalies

observed in the Vostok ice core (Figure 1 and Section 4.2). To test the robustness and sensitivity of our model to uncertainties

and underlying assumptions, we modified several model parameters. First the limitation of late closure trapping was removed,

hence simulating a symmetrical behavior between early and late trapping. The results, displayed in supplementary Figure S7,

show a clear increase in the amplitude of late closure artifacts. In particular, the enhanced late trapping produces artifacts of

about 50 ppbv before the onset of the DO-17 (in the 914 to 917 m depth range). Their absence in the CFA measurements

confirms our assumption of predominance of early closure artifacts. On the other hand, as shown in case B of Figure 3, some

limited late trapping is required to reproduce what appears as positive anomalies at the onset of DO-17 event (912 to 913 m

depth range). We also estimated the sensitivity of the model to the density variability ($\Delta\rho$) and densification rate ($d\rho/dz$).

Extremal values for these two parameters ~~result in characteristic depth shifts~~, provided at the beginning of this section, result

in typical depth anomalies of 1.2 ~~to 4.1 m for the anomalous layers, and~~ and 4.1m, corresponding age anomalies of 99 yr and

341 yr. The model results are displayed in Figures S8 and S9 (SI). Using a reduced depth ~~shift~~ anomaly of the anomalous layers

leads to largely reduced amplitudes of the anomalies. Using an increased depth ~~shift~~ anomaly of the anomalous layers leads to

overestimated amplitudes of the anomalies, especially between 903 and 910 m depth. ~~Note that a reduced snow accumulation~~

~~rate has a similar effect, thus the most arid sites are highly sensitive to layered trapping anomalies.~~ As using a Gaussian

**Table 1.** Layering model parameters and resulting depth anomaly, age anomaly, and associated Figure. The first row corresponds to the reference simulation and sensitivity tests are below. The depth and age anomaly values refer to the standard deviation ($1\sigma$) of early trapping artifacts. These $1\sigma$ values are half the $2\sigma$ values mentioned in the corresponding Figure captions.

| $d\rho/dz$ ($\mathrm{kg\,m^{-4}}$) | $\Delta\rho$ ($\mathrm{kg\,m^{-3}}$) | Limit late anomalies | Narrow GADs | Depth anomaly (m) | Age anomaly (yr) | Figure |
|---|---|---|---|---|---|---|
| 2 | 5 | Yes | No | 2.5 | 207 | 4 |
| 2 | 5 | No | No | 2.5 | 207 | S7 |
| 2.5 | 3 | Yes | No | 1.2 | 99 | S8 |
| 1.7 | 7 | Yes | No | 4.1 | 341 | S9 |
| 2 | 5 | Yes | Yes | 2.5 | 207 | S10 |

distribution of density anomalies is equivalent to using a random depth ~~shift~~anomaly, the smallest anomalies produced by the model do not exceed the analytical noise. We imposed a density of 10 anomalies per meter, which results in about 5 significant anomalies per meter (exceeding $10\,\mathrm{ppb}$) in the 895 to $915\,\mathrm{m}$ depth range. About $70\%$ of these significant artifacts correspond to early closure layers. The width of the anomalous layers also influences the amplitude of the modeled anomalies because

it is in the attenuation range of the CFA system. While $2\,\mathrm{cm}$ layers experience a ~~dampening~~ damping of $18\%$, an attenuation of about $30\%$ is observed with $1\,\mathrm{cm}$ layers. The anomalies observed in the Vostok signal have widths ranging between one and a few centimeters. Their smoothing by the CFA system is thus limited. We also tested an alternative to the homogeneous GADs hypothesis, assuming that anomalous layers have a strongly reduced GAD: similar to the gas age distribution in the ~~WAIS~~ WDC core. The results are displayed in Figure S10 (SI). As the ~~WAIS~~ WDC record of DO-17 event is less smooth than

the Vostok record, the reduced GAD assumption leads to large positive artifacts, especially around $912\,\mathrm{m}$ depth, which are not observed in the Vostok signal. ~~Finally,~~

Finally, under the hypothesis of density based layering, age anomalies strongly depend on accumulation as explained by Rhodes et al. (2016) . A lower accumulation leads to a weaker density variability in the firn (Hörhold et al., 2011) , but at the same time leads to a larger age difference between successive firn layers due to a steeper age-depth slope. The second

effect tends to dominate and the net effect of a lower accumulation is an increase in age anomalies due to layered trapping. Moreover, it is important to note that the good agreement between our density driven model and observations does not imply that tortuosity is not an important factor in anomalous trapping. High resolution air content measurements could potentially help better understanding the physical properties of anomalous layers at closure time.

## 4.4 Removing layering artifacts in the methane record

To extract an undisturbed (chronologically monotonous and representative of atmospheric variability only) methane signal from the Vostok 4G-2 core, layered trapping artifacts need to be removed from the high resolution CFA record. Some sections of the core exhibit mainly positive or negative artifacts. Hence removing them using a running average would bias the signal.

To account for this specificity, a cleaning algorithm has been developed. The underlying assumptions are that the chronological signal is a slowly varying signal with a superimposed noise composed of the analytical noise and of the layered trapping artifacts. ~~The~~ Using a looping procedure, the artifacts are progressively trimmed until the resulting noise is free of spikes. The detailed algorithm is the following:

- Using the CFA signal (~~raw or with~~ with or without already partially removed ~~artifacts~~layering artifacts during the cleaning

process) a running median is computed with a window of $15\,\mathrm{cm}$. Then a binned mean is computed with bins of $50\,\mathrm{cm}$. The goal of this step is to remove noise, without introducing a bias due to layering artifacts.

- A spline of degree 3 is used to interpolate between the binned points on the original CFA depth scale. This ~~spline~~ interpolating spline does not further smooth the signal, and is used as a guess of the chronological signal.

- By removing the spline from the CFA signal we obtain the detrended noise of the signal, composed of the analytical noise

and the remaining artifacts.

- We then compute the Normalized Median Absolute Deviation (NMAD) of the detrended noise. The expression of the NMAD is $1.4826 \times \mathrm{med}(|x_i - \mathrm{med}(x_i)|)$, where $x_i$ are the noise values and $\mathrm{med}$ the median. This is a robust estimator of variability, weakly sensitive to outliers (Höhle and Höhle, 2009; Rousseeuw and Hubert, 2011). It enables to estimate the variability of the noise without the artifacts, that is to say the analytical noise.

- The detrended noise is cut-off with a threshold of $2.5$ times the NMAD.

- We then check if the noise is free of spikes. For this we compare the NMAD (estimation of the variability without spikes) and the Standard Deviation (estimation of variability with spikes) of the detrended noise. If these two quantities are similar, the noise is free of anomalous layers. Here if the Standard Deviation is lower than $1.5$ times the NMAD, the procedure is finished. Otherwise, the algorithm is looped.

This algorithm does not require an estimation of the analytical noise beforehand, since this value is dynamically computed. However, it is sensitive to the value of $1.5$ used to compare the NMAD and Standard Deviation to test the presence of artifacts. The remaining signal after cutting-off the layered trapping anomalies has a noise amplitude of $\pm 16\,\mathrm{ppbv}$, and is represented in black in Figure 1. With our method $15\%$ of the methane data points have been removed. As expected, the signal is almost

not modified below 915m, with a portion of removed points of only $1.3\%$. On the other hand, the variability above $915\,\mathrm{m}$ is greatly reduced and about $26\%$ of the methane data points have been removed.

## 5 Smoothing and age distribution in the Vostok 4G-2 ice core

### 5.1 The smoothing of the methane record

Once the methane signal is cleaned from layered trapping artifacts, we consider that we have access to a chronologically-ordered and unbiased signal recorded in the core. It is smoothed (high frequencies are ~~dampened~~damped) with respect to the true atmospheric signal, and can be used to infer the degree of smoothing in the Vostok ice core. The ~~dampening~~ damping can be visualized in Figure 5 by comparing the Vostok record with the ~~WAIS~~ WDC record. High frequency atmospheric variability is much better preserved in the WAIS Divide ice core because the accumulation rate is more than an order of magnitude higher

(in the range from $18$ to $22\,\mathrm{cm\,ice\,yr^{-1}}$ for the studied period, Buizert et al., 2015) thus the firn densification and gas trapping are faster. For instance, the methane variation spanning between $59,000$ and $58,800\,\mathrm{yrBP}$ is ~~dampened~~damped by $\sim 50\%$ in the Vostok record compared to ~~WAIS~~WDC. Moreover, a $20\,\mathrm{ppbv}$ sub-centennial variation is present in the ~~WAIS~~ WDC record between $58,700$ and $58,600\,\mathrm{yrBP}$. In the Vostok record, however, this short-scale variability event has been smoothed out. On the other hand the multi-centennial variability visible between $58,700$ and $58,400\,\mathrm{yrBP}$ is well preserved with only a slight

damping. From the comparison between ~~WAIS~~ WDC and Vostok, we can infer that the smoothing in Vostok 4G-2 prevents to retrieve information below the centennial scale during the DO-17 period.

### 5.2 Estimate of the gas age distribution

The smoothing of gas concentrations in ice core records is the direct consequence of the broad gas age distributions in the ice (Spahni et al., 2003; Joos and Spahni, 2008; Köhler et al., 2011; Ahn et al., 2014). We call absolute GAD the age distribution

expressed on an absolute time scale, in years before present. The relative GAD is the distribution expressed relatively to its mean age. For a given layer, absolute and relative GAD thus only differ by a translation in age. Here we assume that all layers densified under the same physical conditions, hence sharing the same relative GAD. Since computing concentrations along an ice core using GADs is equivalent to a convolution product (Rommelaere et al., 1997), the resulting concentrations will be called convolved signals.

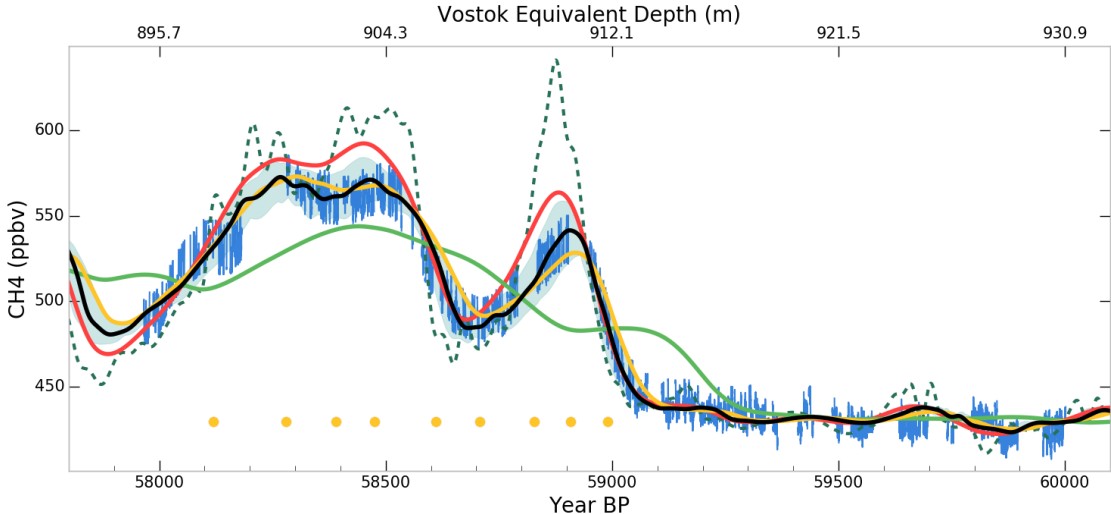

**Figure 5.** DIFdelbegin ~~Methane concentrations observed and calculated using different GADs as a function of gas age and depth. The WAIS record (Rhodes et al., 2015) is displayed in dashed green, and the CFA Vostok measurements in blue. In black: convolved DO-17 methane signal with the optimized GAD from Section 5.2 (uncertainty envelope shown in light blue). In yellow: convolved signal with the modern Vostok GAD from Witrant et al. (2012). In solid green: convolved signal with the Dome C GAD estimated for LGM (Köhler et al., 2011). Yellow dots show the tie points used to match the WAIS and Vostok records.~~WDC $CH_4$ signal convolved with different GADs: the Dome C GAD estimated for the Bølling-Allerød by Köhler et al. (2015) in red, the Dome C GAD estimated for the Last Glacial Maximum by Köhler et al. (2011) in green, the modern Vostok GAD from Witrant et al. (2012) in yellow, and the Vostok DO-17 GAD estimated Section 5.2 in black (uncertainty envelope shown in light blue). The WDC record (Rhodes et al., 2015) is displayed in dashed green, and the CFA Vostok measurements in blue. Yellow dots show the tie points used to match the WDC and Vostok records.

The climatic conditions of the glacial period on the Antarctic plateau have no modern analogue, thus relevant GADs cannot be inferred from modern firn observations. High resolution, CFA based, gas records offer a new opportunity to estimate GADs without modern analogue. We thus developed such a method, which requires a reference atmospheric scenario with much higher frequencies resolved. The method can be extended to other gases than methane or to other low accumulation records than the Vostok 4G-2 core. The principle of the method is to determine a GAD able to convolve the high accumulation record (in our case, WAIS Divide) into a smoothed signal which minimizes the differences with the observed low accumulation record (in our case, Vostok). It can be seen as an inverse problem. Two assumptions are made to reduce the number of adjusted parameters, and thus ensure that the problem is well-defined in a mathematical sense. First all ice layers have the same relative GAD over the considered period. Second, following Köhler et al. (2011), this relative GAD is assumed to be a log-normal distribution, which is fully characterized by two free parameters (for instance its mean and standard deviation). Due to the asymmetry of the GAD, the resulting convolved signal displays age shifts when compared with the original atmospheric scenario. Hence for a valid comparison between the record and convolved signals, it is necessary to modify the age scale and to optimize the GAD in an iterative process. Using an initial age scale, the steps are:

-1: First a new gas age scale is derived. Tie points are manually selected between the low accumulation record and the convolved high accumulation record. The tie points we selected correspond to minima, maxima and mid-slopes points of the methane record. For the initialization, since no GAD has been optimized yet, we use the atmospheric scenario instead of the convolved signal. The new gas chronology is then generated by interpolation and extrapolation between tie points.

-2: A new log-normal GAD is optimized, by modifying its two parameters in order to minimize differences between the simulated and observed smoothed signals. We performed this optimization with a differential evolution algorithm (Storn and Price, 1997).

-3: If five times in a row, the definition of a new chronology and a new GAD does not improve the RMSD (root mean square deviation) between the convolved signal and the measurements, then the algorithm is stopped.

The above methodology can be applied to different ice drilling sites. Here we describe the specific aspects to match the Vostok record with ~~WAIS Divide. The WAIS Divide~~ WDC. Rhodes et al. (2015) state that 'Only at gas ages $> 60$ ka BP is there a possibility that the continuous measurement system caused dampening of the $CH_4$ signal greater than that already imparted by firn-based smoothing processes'. Moreover, Figure S1 of their supplement predicts a GAD width of about $40$ yr for the

DO-17 event, far beyond the width of the Vostok GAD. This ensures that the WDC signal resolves enough high frequencies to be used as the weakly smoothed atmospheric scenario compared to the Vostok record. As explained Section 4.2, the WD2014 gas chronology is converted to the ~~AICC2012~~ GICC05 scale (Buizert et al., 2015) and not further modified. The algorithm only adjusts the Vostok gas ages, which remain well within AICC2012 uncertainties. The initial gas ages used are the ones derived from nitrogen isotopes measurements in Section 3.2, and the optimization has been performed on data ranging from

900 to $915$ m depth. This depth interval has been chosen since it corresponds to a significantly dampened event in the Vostok record, which is sensitive to the choice of the GAD. The optimized gas age distribution is displayed on Figure 6 in black, with uncertainty intervals shown as light blue shaded area. The uncertainty envelope encloses all the distributions resulting in simulated Vostok signals with a RMSD from the measurements lower than $150\%$ of the optimal RMSD. The optimal log-normal parameters are given Table 2. The chosen tie points are displayed in Figure 5, and the optimized $\Delta$Age values along

the Vostok core are depicted in green in Figure 2. The optimal convolution of the ~~WAIS~~ WDC methane record from Rhodes et al. (2015) into a Vostok signal can be seen in black in Figure 5, with the impact of the uncertainty on the GAD displayed as the light blue envelope. The convolution fits the methane measurements within the analytical noise. The overall consistency between the measured and simulated Vostok signals confirms that the Vostok record is a smoothed version of the ~~WAIS~~ WDC record, and that the choice of a single GAD for the whole DO-17 record is a credible hypothesis. This last point is consistent

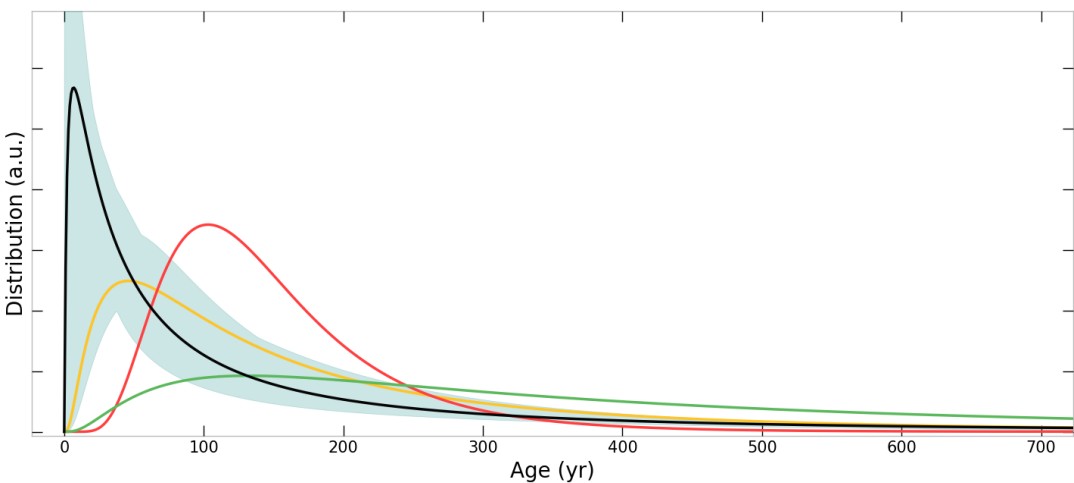

**Figure 6.** Gas Age Distributions. In black: the Vostok GAD during the DO-17 estimated with our optimization scheme, the uncertainty envelope is shown in light blue. In yellow: the modern conditions Vostok GAD estimate from Witrant et al. (2012). In red: the estimated Dome C GAD during B/A from Köhler et al. (2015) . In green: the estimated Dome C GAD during LGM from Köhler et al. (2011).

with the fairly stable climatic conditions on the Antarctic plateau over this time period (Petit et al., 1999; Bazin et al., 2013; Veres et al., 2013).

## 6 Discussion

### 6.1 Understanding the smoothing of ice core signals under low accumulation conditions

5   In Figure 6, our GAD adjusted to produce the expected smoothing rate for the DO-17 event in the Vostok ice core (in black) is compared to other available gas age distributions for low accumulation rate conditions. The different parameters of the lognormal GADs used in this section are displayed Table 2. For modern ice cores, GADs can be estimated with gas transport models constrained by firn air composition data (Buizert et al., 2012; Witrant et al., 2012). However, the results directly depend on the closed versus total porosity parameterization used, which is insufficiently constrained (e.g. Mitchell et al., 2015).

10  A direct comparison of our optimized GAD for Vostok during DO-17 and a GAD constrained with modern condition firn-air measurements at Vostok ~~(Witrant et al., 2012) (in yellow in Figure 6)~~ (Witrant et al., 2012, in yellow in Figure 6) suggests a slightly narrower distribution for the glacial period, despite lower temperatures. ~~On the other hand, the GAD estimate from Köhler et al. (2011) for Dome C during the Last Glacial Maximum (LGM) is much wider (in green) and results in a too smoothed Vostok methane record in Figure 5. The GADs calculated for modern conditions from Köhler et al. (2011) at~~

**Table 2.** Parameters defining the log-normal distribution used as GADs, for Vostok DO-17 (this study), modern Vostok (Witrant et al., 2012), Dome C during the Bølling-Allerød (Dome C B/A, Köhler et al., 2015), and Dome C during the Last Glacial Maximum (Dome C LGM, Köhler et al., 2011). Location and scale respectively refer to the parameters $\mu$ and $\sigma$ used in Equation 1 in Köhler et al. (2011). Std Dev stands for Standard Deviation.

| Site and Period | Location | Scale | Mean (yr) | Std Dev (yr) |
|---|---|---|---|---|
| Vostok DO-17 | 4.337 | 1.561 | 259 | 835 |
| Vostok Modern | 4.886 | 1.029 | 226 | 308 |
| Dome C B/A | 4.886 | 0.5 | 150 | 79 |
| Dome C LGM | 5.880 | 1 | 590 | 773 |

~~Dome C and Witrant et al. (2012) at Vostok are very similar, which is consistent with the comparable accumulation rates of the two sites: 2.7 cm ice yr$^{-1}$ at Dome C (Gautier et al., 2016) and 2.4 cm ice yr$^{-1}$ at Vostok (Arnaud et al., 2000). Thus a wider GAD, as obtained by Köhler et al. (2011), is expected under the much drier conditions of the DO-17 event (1.3 instead of 2.4 cm ice yr$^{-1}$). However, convolving the WAIS signal with this LGM distribution during the DO-17 event leads to in a~~

5  ~~simulated methane signal much smoother than experimentally observed along the Vostok record.~~ On the other hand, the GAD estimate from Köhler et al. (2015) for Dome C during Bølling-Allerød (B/A, accumulation of about 1.5 cm ice yr$^{-1}$) is narrower (in red Figure 6) and results in a slightly too weakly smoothed methane record in Figure 5. Finally, the GAD proposed by Köhler et al. (2011) for Dome C during the Last Glacial Maximum (LGM) is broader than the other presented GADs (in green Figure 6), and thus leads to a stronger smoothing in the record Figure 5. The GADs calculated for modern conditions

10  from Köhler et al. (2011) at Dome C and Witrant et al. (2012) at Vostok are very similar, which is consistent with the comparable accumulation rates of the two sites: 2.7 cm ice yr$^{-1}$ at Dome C (Gautier et al., 2016) and 2.4 cm ice yr$^{-1}$ at Vostok (Arnaud et al., 2000). We therefore do not observe a systematic broadening of GADs for lower accumulation rates, even at a given site. It questions either the relationship between GAD widths and accumulation rate, or the consistency between GADs derived from gas transport models in firn and the GAD obtained with our method of record comparison.

The most likely reason for an inconsistency between GADs inferred from firn models and from CFA data is the large uncertainty on the representation of gas trapping in firn models. As mentioned above, the closed versus total porosity ratio is very uncertain, as it was measured only at a few sites and on small size samples. Better constraints on the physics of gas trapping would thus be helpful. However, there is no modern analogue of the central Antarctic plateau sites (such as Vostok or Dome C) under glacial conditions. Thus using CFA high resolution gas measurements at different sites to constrain Holocene GADs at

low accumulation sites would be the only way to check the consistency of the two methods. Previous comparisons between sites indicate that the smoothing is larger for low accumulation conditions (Spahni et al., 2003; Joos and Spahni, 2008; Köhler et al., 2011; Ahn et al., 2014). Indeed, a simple argument is that the lower the accumulation and the temperature, the slower a firn layer will densify, and thus the broader the GAD. The comparison of the DO-17 records between ~~WAIS~~ WDC and Vostok 4G-2

corroborates this relationship: the higher accumulation ~~WAIS~~ WDC signal is less smooth than the Vostok signal (Figure 5). ~~Moreover, the impact of layering on the overall gas age distribution is unknown, and the Vostok record of~~ The weaker than expected smoothing during DO-17 ~~event strongly suggests an important layering effect even in very arid conditions~~ at Vostok could be due to the presence of a strong layering preventing air renewal and mixing, as suggested in Mitchell et al. (2015) .

From a paleo-climatic point of view, an important conclusion of this work is that the smoothing of atmospheric trace gases recorded in ice cores from the central Antarctic plateau could be less important than expected under glacial conditions, resulting in more retrievable information about past atmospheric conditions. Ice cores with the oldest enclosed gases, such as in the Oldest Ice project (Fischer et al., 2013), will be retrieved from very low accumulation sites. They could thus potentially provide meaningful information down to the multi-centennial scale.

**6.2  Layered trapping and atmospheric trend reconstructions**

The anomalous layers in the Vostok methane record discussed Section 4.2 are one to a few centimeters thick, and discrete samples used for methane measurements in ice cores are typically also a few centimeters thick. In our study, the use of high resolution continuous analysis made it possible to identify abnormal methane values which appeared as spikes in the record. However, in the case of discrete measurements, the absence of continuous information makes it hard to discriminate between

normal and abnormal layers. For instance, the comparison of the ~~WAIS~~ WDC continuous record and the EPICA Dome C (EDC) discrete methane record (Loulergue et al., 2008) indicates a potential artifact during the onset of the Dansgaard-Oeschger event 8 ($\sim 38,000\,\mathrm{yrBP}$), as displayed in Figure 7. One of the EDC samples shows a reduced methane concentration which should be visible in the less smooth ~~WAIS~~ WDC record as well, if this corresponded to a true atmospheric feature. Moreover, the measured mixing ratio in this EDC sample is consistent with an artifact resulting from early gas trapping. As mentioned in Rhodes

et al. (2016), and as confirmed by our study, it is important for paleoclimatic studies to avoid interpreting such abnormal values as fast atmospheric events.

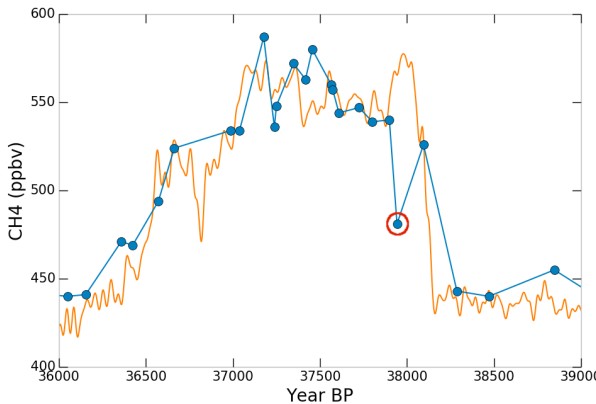

**Figure 7.** Discrete EDC methane record (blue) and continuous ~~WAIS~~ WDC methane record (orange). The ~~WAIS~~ WDC record was put on the ~~AICC2012~~ GICC05 time scale, and then shifted by 250 yr to improve matching. We suggest that the circled point corresponds to a layered trapping artifact.

However, continuous flow analysis may not always allow ~~distinguishing between anomalous layers~~ us to distinguish between layering artifacts and the chronologically ordered signal. The deep parts of ice cores with low accumulation and high thinning are of particular interest in paleoclimatology since they enclose very old gases (Loulergue et al., 2008; Lüthi et al., 2008). However, with a strong thinning, the width of abnormal layers may shrink below the spatial resolution limit of analytical

5   systems. In such a case, an average mixing ratio over several layers is measured. Since layered trapping artifacts are unevenly distributed in term of sign, they bias the measured average signal. ~~For instance, for a record with artifacts similar to the DO-17 in Vostok 4G-2, covering about 15% of the core and reaching 50 ppbv, this bias is about 7 ppbv~~ In the very simple case of a record with artifacts all negatively orientated, covering $15\%$ of the ice core and all reaching $50$ ppbv, this bias is about $-7$ ppbv. In the case of records with lower accumulation or stronger methane variations the bias will be even more important.

10  The development of very high resolution gas measurement techniques, thus offers important perspectives for analyzing the deepest part of ice cores. In intermediate situations where anomalous layers could be distinguished but a high accumulation record is not available (before the last glacial-interglacial cycle), the effect of smoothing is more difficult to constrain but the presence of layered trapping ~~artifact~~ artifacts is in itself an indication that some smoothing may occur because layered trapping occurs only in fast atmospheric change conditions.

# 7 Conclusions

We presented the first very high resolution record of methane in an ice core sequence formed under very low accumulation rate conditions. It covers the gas record of Dansgaard-Oeschger event 17, chosen for its abrupt atmospheric methane changes at a similar time scale as gas trapping.

The continuous flow analysis system, optimized to reduce gas mixing, allowed us to reveal numerous centimeter scale methane concentration anomalies. Positive anomalies affecting both the methane and carbon monoxide records were attributed to kerosene contamination and discarded. The remaining anomalies are unevenly distributed, a few centimeters wide and mostly negatively oriented with dips as low as $-50\mathrm{ppbv}$. The anomalies occur only during time periods of fast atmospheric methane variability. The main characteristics of the size and distribution of the anomalies could be reproduced with a simple model based on relating realistic firn density anomalies to early or (to a lesser extent) late trapping. Such layered trapping anomalies may be confused with the climatic signal in discrete climate records or bias the signal if too narrow to be detected by a CFA system (e.g. in the high thinning conditions of the deep part of ice cores). It is important for future paleoclimatic ~~studied~~ studies not to interpret those abrupt variations as fast chronologically-ordered atmospheric variations. Further use of high resolution continuous analysis will allow ~~discriminating~~ us to discriminate for layered trapping artifacts and to better identify their statistical characteristics. Moreover, the sign of the trapping artifacts is not random: some sections of the record display only positive or negative artifacts. Thus simple averaging would result in a systematic bias of the signal. Hence, we developed a cleaning algorithm aiming at minimizing this bias.

After removing the centimeter scale anomalies, the remaining Vostok methane signal is distinctly smoother than the ~~WAIS~~ WDC record (Rhodes et al., 2015). The snow accumulation rate being more than one order of magnitude higher at ~~WAIS~~ WDC than at Vostok, the ~~WAIS~~ WDC signal contains higher frequency features. The comparison of the two signals opens the possibility to estimate gas age distributions for conditions of the East Antarctic plateau during the last glacial period, which have no modern analogue. For the DO-17 event at Vostok, the resulting gas age distribution is narrower than expected from ~~firn models~~ a comparison with modern firns (Köhler et al., 2011; Witrant et al., 2012). It may be due to an incorrect prediction of gas trapping by firn models and/or an incorrect extrapolation of the firn behavior to very low temperature and accumulation conditions. The apparently similar smoothing at Vostok under DO-17 and present conditions contradicts the expected primary effect of temperature and accumulation rate: lower temperature and accumulation rates induce a longer gas trapping duration and thus a stronger smoothing. On the other hand, ~~gas trapping processes are still weakly constrained in firn~~

~~models (e.g. Mitchell et al., 2015)~~ Mitchell et al. (2015) point out the lack of firn layering representation in most firn models and conclude that firn layering narrows gas age distribution in ice. From a paleoclimatic point of view, ice cores with the lowest accumulations contain very old gases. The less important than expected smoothing under glacial conditions implies that atmospheric information at shorter time scale than previously expected might be retrieved. However similar measurements need to be performed on other low accumulation records, to confirm ~~or infirm~~ our results for different sites and/or periods. For the DO-17 event at Vostok, multi-centennial atmospheric variations are still accessible in the record. Further comparisons of high and low accumulation records of the last glacial cycle will allow ~~better constraining~~ us to better constrain the relationship between ice cores and atmospheric gas signals, even for no modern analogue conditions.

## 8 Code availability

Numerical codes were developed using Python 2.7 and readily-available packages (numpy, scipy, etc). They will be provided upon direct request to the corresponding authors.

## 9 Data availability

Datasets produced during this study will be made available in the World Data Center for Paleoclimatology (WDC Paleo).

*Author contributions.* Methane measurements were carried out by Xavier Faïn and Kévin Fourteau. Nitrogen isotopes measurements were carried out by Kévin Fourteau and Amaëlle Landais. Numerical codes were designed and developed by Kévin Fourteau and Patricia Martinerie. The ice core samples were made available thank to Vladimir Ya. Lipenkov and Jérôme Chappellaz. They were cut and sent to France from Vostok station by Alexey A. Ekaykin. All co-authors contributed to the data analysis and interpretation. The manuscript was written by

5    Kévin Fourteau with the help of all co-authors.

*Competing interests.* The authors declare that they have no conflict of interest.

*Acknowledgements.* The research leading to these results has received funding from the European Community's Seventh Framework Programme ERC2011 under grant agreement No. 291062 (ERC Ice&Lasers), the INSU/CNRS LEFE project NEVE-CLIMAT, the Laboratoire International Associé Vostok partnership, the LabEx OSUG@2020 (Investissements d'avenir - ANR10LABX56), and the École Normale

10   Supérieure Paris-Saclay. Ice core samples were made available within the Laboratoire International Associé Vostok partnership. We are grateful to the Russian Antarctic Expeditions for carrying out the logistics and the shipping of the ice core samples to Europe. We also thank Frédéric Prié for his help during nitrogen isotopes measurements, and Maurine Montagnat and Cédric Lachaud for their help with the thin sections. Finally, we thank Hubertus Fischer and the two anonymous referees for their constructive and helpful comments on this article.

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
