# Peer review of "Analytical constraints on layered gas trapping and smoothing of atmospheric variability in ice under low accumulation conditions"

_Climate of the Past, 2017_

## Referee Comment (RC1) · Anonymous Referee #1 · 2 Aug 2017

This paper analyses $CH_4$ across Dansgaard/Oeschger event 17 (DO-17) from the Vostok ice core with a CFA-based measurement system in order to improve understanding of layered gas trapping and smoothing of atmospheric variability in an ice core drilled in low accumulation areas. A thus dervied $CH_4$ record is then postprocessed and finally compared with $CH_4$ from the higher resolving WAIS Divide ice core (WDC) to conclude that gas age distribution (GAD) - or smoothing - in Vostok seems to be similar for modern and DO-17 conditions.

The paper is well written, especially the post-processing procedure described to

some detail. The detection of artifacts in $CH_4$ from such a high-resolution system as presented here is convincing.

However, the paper falls so far short in one aspect, that is the application of a previously assumed LGM gas age distribution used for EPICA Dome C (EDC) to transfer WDC $CH_4$ data into potentially signals recorded in Vostok, from which it was concluded, that gas age distribution are probably independent from climate background. Here, they use what has been used as gas age distribution in Köhler et al (2011), who used a log-normal function, and for LGM assumed a mean width of the GAD of 590 years. This GAD allowed large overshoots in the true atmospheric signal of $CO_2$, when compared with the EDC ice core record of $CO_2$, and was again used in Köhler et al. (2014). However, the new WDC $CO_2$ paper of Marcott et al (2014) showed, that the assumed GAD used by Köhler in 2011 was probably too wide since a much smaller GAD was able to transfer the WDC $CO_2$ (potentially very close to the true atmospheric variability of $CO_2$) to the $CO_2$ record obtained from EDC (Extended Data Fig 5 in Marcott et al., 2014). This revised narrow GAD was also then applied for the question of interest by Köhler, but I have to admit, so far only published in a conference proceeding, not widely known (Köhler et al., 2015, pages 135–140 in http://www.leopoldina.org/uploads/tx_leopublication/NAL_Bd121_Nr408_LR.pdf). Figure 2 of this 6-pages proceeding contains a transformation of a simulated atmospheric $CO_2$ into a signal recorded at EDC around the onset of the Bølling/Allerød (B/A) warm period around 14.6 kyr BP using the same log-normal function as introduced in Köhler et al (2011) of

$$y = \frac{1}{x \cdot \sigma \cdot \sqrt{2\pi}} \cdot e^{-0.5(\frac{ln(x)-\mu}{\sigma})^2} \tag{1}$$

with $x$ (yr) as the time elapsed since the last exchange with the atmosphere, which leads to an *expected value (mean)* $E$ of the GAD of

$$E = e^{\mu + \frac{\sigma}{2}}. \tag{2}$$

From the two free parameters $\mu$ and $\sigma$, in 2011 Köhler has chosen for simplicity $\sigma = 1$, but now in the revised application in 2015 uses $\sigma < 1$ to reproduce the shape of the GAD suggested in Marcott. In detail $\sigma = 0.5$ was used and $\mu$ defined in a way which guarantees the pre-defined mean values $E$ of 150 yr. So, not only the mean width of the GAD has be reduced by a factor of 2.7, from formally 400 yrs to now 150 yrs (for this application to the B/A), but also the shape of the GAD.

I believe the authors are challenged now to also use a GAD that agrees with the WDC-EDC $CO_2$ comparsion, as brought up by Marcott, and in a first step probably at best also start with a revised log-normal function using different parameter values. Only then can they conclude (or not) if the GAD is indeed similar for modern and DO-17 climate or not. For any such exercise, please always state the used parameter values of the function, e.g. as given here, both the chosen form-shape factor $\sigma$, and at best the mean value $E$ (directly derived from $\mu$ once $\sigma$ is given). So far, no details on the applied log-normal function has been given. This will probably lead to a revision of the final conclusion and figure 5, but the rest of the paper is largely unaffected.

**Further minor comments in chronological order:**

1. Throughout the paper: Units are sometimes weight, with a dot (.) inbetween, e.g. "3.8 cm.min$^{-1}$", which should be "3.8 cm min$^{-1}$".

2. Figure 1: Labels in insert (top right corner) are much too small.

3. Page 9, line 4: " As explained in Rhodes et al. (2016), such a mechanism affects trace gases record only during periods of significant atmospheric variations." Variations of what? Probably "variations in concentrations of atmospheric gases".

4. Page 9, line 11: "monotonous variations", change to "monotonous in/decrease".

5. Page 11, line 21: What are the coldest sites in Breant et al (2016)? Please name here.

6. Page 12, line 4: "... the methane record from the WAIS Divide ice core (Rhodes et al., 2015), with gas ages with gas ages converted on the AICC2012 scale (Buizert et al., 2015)." Now, this needs some more explanation and probably correction. Buizert et al., 2015 does NOT plot WDC $CH_4$ on AICC2012, as suggested by this sentence. There is also the effort of explaining the gas age adaption of WDC $CH_4$ to AICC2012 in the SI Fig S11 (and corresponding SI section), which I also did not understand in detail. Please be precise here, and describe this step in the main text, not hidden in the SI.

7. Page 14, line 3: I believe, a spline normally comes along with a cutoff-frequency, which has not been given here.

8. Section 4.4 (removing artifacts) page 13-14, versus Fig 1. My understanding of the description of Section 4.4 was, that the spikes caused by layering artifacts are removed, and a continous $CH_4$ time series without artifacts is generated. However, the black line in Fig 1 (which according to the text should be such a time series) does not contain any data in the periods, in which artifacts has been removed. I would think the post-processing should give you some data points in exchange to the removed artifical peaks. Please refine text, or change Figure 1 accordingly.

9. Caption to Fig 5: You need to say explicitly WHICH signals you convoluted, e.g. green solid is probably the convolution of the WDC $CH_4$ with the Dome C GAD estimated for LGM of Köhler et al 2011.

10. Page 16, line 20: "modifying its two parameters", probably refers to the same 2 parameters given above in Eq 1. Please state, which values you choose in the end.

11. Fig 6: Needs a new GAD based on the Marcott WDC-EDC $CO_2$ comparison, and/or the new approach of Köhler 2015.

12. SI: Please either put all Figures to the end, or in the section, in which they are discussed.

13. Please check references to Figures in main text, on SI page 5, line 4 a reference is given to Figre 6, but the correct Figure refered to here is Figure 5.

**References**

Köhler, P; Knorr, G.; Buiron, D.; Lourantou, A.; Chappellaz, J. Abrupt rise in atmospheric o at the onset of the Bølling/Allerød: in-situ ice core data versus true atmospheric signals, Climate of the Past, 2011, 7, 473-486

Köhler, P.; Knorr, G.; Bard, E. Permafrost thawing as a possible source of abrupt carbon release at the onset of the Bølling/Allerød, Nature Communications, 2014, 5, 5520.

Köhler, P.; Völker, C.; Knorr, G.; Bard, E. High Latitude Impacts on Deglacial $CO_2$: Southern Ocean Westerly Winds and Northern Hemisphere Permafrost, Thawing Nova Acta Leopoldina, Leopoldina, 2015, 121, 135-140.

Marcott, S. A.; Bauska, T. K.; Buizert, C.; Steig, E. J.; Rosen, J. L.; Cuffey, K. M.; Fudge, T. J.; Severinghaus, J. P.; Ahn, J.; Kalk, M. L.; McConnell, J. R.; Sowers, T.; Taylor, K. C.; White, J. W.; Brook, E. J. Centennial Scale Changes in the Global Carbon Cycle During the Last Deglaciation, Nature, 2014, 514, 616-619

---

## Referee Comment (RC2) · Anonymous Referee #2 · 5 Aug 2017

This study presents a high quality, novel data set consisting of ultra-high resolution methane measurements across Dansgaard-Oeschger event 17 in the low accumulation Vostok ice core from East Antarctica. The incredible detail of this record reveals rapid, anomalous signals that do not reflect past atmospheric changes, but are instead related to the process of time-varying gas trapping in the firn column. The authors develop a simple but effective numerical model to simulate the formation of these gas trapping artifacts, facilitating their removal and obtainment of a solely atmospheric signal. The Vostok atmospheric signal contains more high frequency information than would be expected from existing firn model-based predictions. A revised, much narrower, estimate of the gas age distribution at Vostok is produced. Although more work

is needed to confirm these findings, the implication is that more detailed atmospheric records can be obtained from the older ice located in the Antarctic interior.

The paper will be of interest to many in the ice core community and its implications are particularly relevant for the future development of CFA gas measurements and the search for the oldest ice. It is well written with excellent figures. I include many comments, but they should be straightforward for the authors to address.

Understanding gas trapping Can any more information be provided about the frequency of the gas trapping artifacts in depth and ice age domain? The signals reported by Rhodes et al. 2016 were annual – is the variability closer to decadal here and what does this suggest about the physical heterogeneity responsible? Is any comparison with high resolution chemistry possible across this interval?

Simple model of layered trapping Section 4.3. - Please provide more explanation of how extrapolation of Hörhold data to obtain density variability is carried out. What does the range in density variability obtained represent?

- Pg. 11, lines 20, 22-25: is the "closure depth shift" the difference in depth of pore closure between adjacent layers of different density? Please state clearly if so. And is the "age shift" of 207 yrs equivalent to the age difference between the gas trapped in adjacent layers? Does this value change as a result of subsequent tests? Pg. 13, line 4: are "age anomalies" the same as "age shift"?

- A series of tests are conducted to illustrate the sensitivity of the model to input parameters. This is important and interesting but not that clear. Add some paragraphs please. Line 3 – make it clear that the extreme values used are the max and min of ranges already stated. A table including the parameters used and the resulting age and depth shifts would be informative.

- Can anything be said about the relative importance of accumulation rate and density variability in causing gas trapping? Sites like Vostok have low accumulation, causing

higher CH4 anomalies than high accumulation sites, but cold, low accumulation sites also tend to have lower density variability at depth (Fig. 7F, Horhold et al. 2012), which would cause lower CH4 anomalies.

Estimation of Vostok GAD Section 5.2 - Needs an existing high(er) resolution CH4 record. No record exists beyond ~100 ka (NEEM), which limits application of this technique. Abstract (line 14) should be modified to state need for higher resolution record. Still, it will be really interesting to see method applied to other sites for the Last Glacial.

- Pg. 17, line 10-11. WD also experiences stable conditions over this time period. What if the reference atmospheric scenario was from NEEM where accumulation and temperature change greatly across DO events? Would method need to be adapted?

- Related to this, how valid is the assumption that WD represents the atmosphere? Why isn't this record also biased by gas trapping effects (high accumulation so faster trapping? more CFA smoothing?)?

- Pg. 19, line 2. Is the impact of layering on GAD really "unknown". Mitchell et al. (2015) state "total net effect of layering on gas trapping and the width of the age distribution of gases are unquestionably to narrow the age distribution" and your results seem to support this. Some discussion of the modelling work in Mitchell et al. (2015) might help the discussion here.

Also on Pg. 21, line 18 – a sentence or two summarizing the findings of Mitchell et al. (2015) would help will the argument that firn models currently do a poor job (or don't attempt) at dealing with layered gas trapping.

Specific comments: Pg. 2, Lines 3-10: Consider stating that this is the 'traditional' description of the firn column. There is evidence, including the gas trapping anomalies presented here, that contradict the idea of bubble closure only occurring in the lock-in zone.

Pg. 3, line 27: Clathrate relaxation cavities are not mentioned again until Pg. 10. line 15. Sentence 'samples showed small clathrate relaxation cavities, the CFA sticks did not reveal visual anomalies'. Isn't this statement contradictory? In which direction would clathrate relaxation affect the CH4 signal and why?

Pg. 3, line 7: 'in periods of fast atmospheric variations...' Be clearer about what this means. Atmospheric variation must occur over the time frame of the gas trapping process (not seasonal variability for example), which will change with ice core analyzed.

line 10: WAIS Divide information in Rhodes et al. (2016) is from a model only.

Pg. 9 & Fig. 3: Great figure. Could an arrow be added to indicate the direction of time (right to left)? In discussion about relative influence of early and late closure on final signal, do you mean proportion of early trapped layers will be greater than later trapped layers? Or, do you mean the amplitude of the early layer signals will be greater than the later layer signals? Could this hypothesis be illustrated on the figure?

Pg. 12 & Pg. 16: Buizert et al. (2012) does not convert gas ages to AICC2012. Do you mean GICC05 here?

Figure 5 & Pg. 15: Yes, the tiny sub-centennial variation is smoothed out in Vostok, but multi-centennial information is preserved, e.g., feature 58.7-58.4 ka. This is more detail than we would expect from Dome C GAD estimation and worth mentioning. It would help justify statement on Pg. 19, line 9 that at the moment is tenuous.

Pg. 20, line 1: Can you be more specific about the 'bias' possibly introduced by gas trapping artifacts? Does the 7 ppbv refer to a positive or negative bias? Wouldn't the direction of bias change with the atmospheric trend and so even itself out over the relatively short timescales of gas trapping (compared to length of record compressed within small depth of ice)?

Supplement, Pg. 1, line 13: statement about WAIS data being scaled to discrete measurements is not accurate.

Technical notes: Pg. 1, Line 13: Add "numerical" method.

Pg. 1, Line 21 and repeatedly through manuscript: "gases get enclosed within bubbles...and allow reconstructing..." "Allow reconstructing" is not grammatically correct and should be changed to something like "allow us to reconstruct..." or "allowing reconstruction of...".

Pg. 2, line 11: change "atmospheric composition events" to "atmospheric variability"

Pg. 2 line 12: "dampening" should read "damping" = the decrease in the amplitude of an oscillation or wave motion with time.

Pg. 3, line 4: define or explain "short scale physical variability"

Pg. 3, line 5: insert "physical" before heterogeneities

Pg. 4, line 2-3 repeats what is said on previous page

Pg. 4, line 12: state volume of debubbler

Pg. 6, line 5 onwards: separate into two paragraphs

Pg. 6, line 15: does 2.1m represent one instance of kerosene contamination or it is the sum of many?

Fig. S3: Add indication of depth range represented.

Fig. 1: increase sub-figure size

Fig. S11 caption and elsewhere: be specific - 'WD2014' gas chronology

Pg. 6, Line 21: change to "atmosphere relevant" of atmospheric history relevant?

Pg. 6, line 28: 50 ppbv amplitude and 2 cm wavelength

Pg. 8, line 26: 'the closure of such a layer is likely progressive' – please clarify meaning

Pg. 9, line 4: clarify 'significant atmospheric variations'. Again, quantify.

Pg. 11, line 9: replace 'later' with 'latter'

Pg. 13, line 21: a signal that is representative of only atmospheric variability

Pg. 13, line 28: be specific here, artifacts already removed are due to breaks or kerosene.

Pg. 14, line 3: provide details on spline fit

Pg. 14, line 27: 'high frequency atmospheric variability'

Pg. 19, line 23: be specific here, "anomalous layers" are 'gas trapping artifacts' or artifacts

also due to other things like kerosene?

Pg. 21, line 21: delete "or infirm"

Strictly, the WAIS Divide ice core should be referred to as WD, not WAIS (the ice sheet).

---

## Editor Comment (EC1) · H. Fischer (Editor) · 14 Sep 2017

The discussion period of your paper has been closed two weeks ago and you are expected to provide a reply to the review comments until the 27th of September. Based on this, I will make the final editor decision.

As the reviews are generally very positive, I do not see major problems for you to respond to the reviewer comments and meet their constructive points for improvements.

What appears important is that you also use a narrower age distribution than the older

work by Koehler et al., which is overly broad, for a final assessment of your layer trapping process compared to classical age distributions. Also a discussion of previous work (for example Mitchell et al.) would corroborate your results.

Looking forward to your replies

Cheers Hubertus

---

## Author Comment (AC1) · 27 Sep 2017

**RESPONSE TO REFEREE 1:**

Please find below the reponses to the review of referee 1 on *Analytical constraints on layered gas trapping and smoothing of atmospheric variability in ice under low accumulation conditions.* The blue italic text is the text of the review, and the corresponding responses are below in black. When we intend to change the manuscript text or figures, it is stated so in the response.

*This paper analyses CH4 across Dansgaard/Oeschger event 17 (DO-17) from the Vostok ice core with a CFA-based measurement system in order to improve understanding of layered gas trapping and smoothing of atmospheric variability in an ice core drilled in low accumulation areas. A thus dervied CH4 record is then postprocessed and finally compared with CH4 from the higher resolving WAIS Divide ice core (WDC) to conclude that gas age distribution (GAD) - or smoothing - in Vostok seems to be similar for modern and DO-17 conditions. The paper is well written, especially the post-processing procedure described to some detail. The detection of artifacts in CH4 from such a high-resolution system as presented here is convincing.*

*However, the paper falls so far short in one aspect, that is the application of a previously assumed LGM gas age distribution used for EPICA Dome C (EDC) to transfer WDC CH4 data into potentially signals recorded in Vostok, from which it was concluded, that gas age distribution are probably independent from climate background.*

Our conclusion that smoothing during DO17 is weaker than expected is mostly based on the comparison between the gas age distribution (GAD) tuned to our $CH_4$ data and the GAD estimated by Witrant et al., 2012 for modern conditions at Vostok.

*Here, they use what has been used as gas age distribution in Köhler et al (2011), who used a log-normal function, and for LGM assumed a mean width of the GAD of 590 years. This GAD allowed large overshoots in the true atmospheric signal of CO2, when compared with the EDC ice core record of CO2, and was again used in Köhler et al. (2014). However, the new WDC CO2 paper of Marcott et al (2014) showed, that the assumed GAD used by Köhler in 2011 was probably too wide since a much smaller GAD was able to transfer the WDC CO2 (potentially very close to the true atmospheric variability of CO2) to the CO2 record obtained from EDC (Extended Data Fig 5 in Marcott et al., 2014). This revised narrow GAD was also then applied for the question of interest by Köhler, but I have to admit, so far only published in a conference proceeding, not widely known (Köhler et al., 2015, pages 135–140 in http://www.leopoldina.org/uploads/tx_leopublication/NAL_Bd121_Nr408_LR.pdf). Figure 2 of this 6-pages proceeding contains a transformation of a simulated atmospheric CO2 into a signal recorded at EDC around the onset of the Bølling/Allerød (B/A) warm period around 14.6 kyr BP using the same log-normal function as introduced in Köhler et al (2011) of*

$$y = \frac{1}{x \cdot \sigma \cdot \sqrt{2\pi}} \cdot e^{-0.5(\frac{ln(x)-\mu}{\sigma})^2} \qquad (1)$$

*with x (yr) as the time elapsed since the last exchange with the atmosphere, which leads to an expected value (mean) E of the GAD of*

$$E = e^{\mu + \frac{\sigma}{2}}. \qquad (2)$$

*From the two free parameters μ and σ, in 2011 Köhler has chosen for simplicity σ = 1, but now in*

*the revised application in 2015 uses σ < 1 to reproduce the shape of the GAD suggested in Marcott. In detail σ = 0.5 was used and μ defined in a way which guarantees the pre-defined mean values E of 150 yr. So, not only the mean width of the GAD has be reduced by a factor of 2.7, from formally 400 yrs to now 150 yrs (for this application to the B/A), but also the shape of the GAD. I believe the authors are challenged now to also use a GAD that agrees with the WDC-EDC CO2 comparsion, as brought up by Marcott, and in a first step probably at best also start with a revised log-normal function using different parameter values. Only then can they conclude (or not) if the GAD is indeed similar for modern and DO-17 climate or not. For any such exercise, please always state the used parameter values of the function, e.g. as given here, both the chosen form-shape factor σ, and at best the mean value E (directly derived from μ once σ is given). So far, no details on the applied log-normal function has been given. This will probably lead to a revision of the final conclusion and figure 5, but the rest of the paper is largely unaffected.*

We will add the new GAD by Köhler et al., 2015 in Figures 5 and 6. This new GAD leads to a somewhat lower smoothing than the modern one based on Witrant et al. (2012). But there are large uncertainties in GAD estimation as we mention p18 l20-25 of the manuscript. However, the main conclusion is unaffected. The smoothing during DO17 is comparable to the modern one at Vostok, despite the lower accumulation.

We will also add a Table with the parameters of all the GADs used to facilitate the comparison with other estimates. Note that we did not assume $\sigma = 1$ but adjusted this parameter in our estimates.

*Further minor comments in chronological order:*

*1. Throughout the paper: Units are sometimes weight, with a dot (.) inbetween, e.g. "3.8 cm.min−1 ", which should be "3.8 cm min−1 ".*

We will remove the dot in between units.

*2. Figure 1: Labels in insert (top right corner) are much too small.*

Figure 1 will be redrawn with larger zoom and labels.

*3. Page 9, line 4: " As explained in Rhodes et al. (2016), such a mechanism affects trace gases record only during periods of significant atmospheric variations." Variations of what? Probably "variations in concentrations of atmospheric gases".*

Yes we meant variations in concentrations of atmospheric gases. The text will be modified accordingly to 'during periods of variations in concentration of atmospheric gases'.

*4. Page 9, line 11: "monotonous variations", change to "monotonous in/decrease".*

The text will be changed to 'monotonous increase/decrease'.

*5. Page 11, line 21: What are the coldest sites in Breant et al (2016)? Please name here.*

The sites are Dome C, Vostok, and Dome A. This will be added to the text.

*6. Page 12, line 4: "... the methane record from the WAIS Divide ice core (Rhodes et al., 2015), with gas ages with gas ages converted on the AICC2012 scale (Buizert et al., 2015)." Now, this needs some more explanation and probably correction. Buizert et al., 2015 does NOT plot WDC CH4 on AICC2012, as suggested by this sentence. There is also the effort of explaining the gas age adaption of WDC CH4 to AICC2012 in the SI Fig S11 (and corresponding SI section), which I also*

did not understand in detail. Please be precise here, and describe this step in the main text, not hidden in the SI.

We will add to the manuscript: 'The WDC gas age chronology (WD2014) was scaled to the GICC05 chronology (with present defined as 1950) dividing by a factor of 1.0063 as in Buizert et al. (2015). For the rest of the article we used this scaled WD2014 chronology to express WDC gas ages.'

*7. Page 14, line 3: I believe, a spline normally comes along with a cutoff-frequency, which has not been given here.*

We are using an interpolating spline, which does not smooth the signal but goes through each data point. The smoothing is taken care of by the averaging over 50cm wide bins, and this averaging length works as the cut-off frequency. The text will be changed to 'A spline of degree 3 is used to interpolate between the binned points on the original CFA depth scale. This interpolating spline does not further smooth the signal, and is used as a guess of the chronological signal.'

*8. Section 4.4 (removing artifacts) page 13-14, versus Fig 1. My understanding of the description of Section 4.4 was, that the spikes caused by layering artifacts are removed, and a continous CH4 time series without artifacts is generated. However, the black line in Fig 1 (which according to the text should be such a time series) does not contain any data in the periods, in which artifacts has been removed. I would think the post-processing should give you some data points in exchange to the removed artifical peaks. Please refine text, or change Figure 1 accordingly.*

Even in the absence of layering artifacts, the CFA signal is discontinuous due to missing ice sections, kerosene and ambient air infiltrations. This fact will be clearly specified in the text. During cleaning of layering artifacts, we do not wish to mix the experimental data with interpolated data that are not independent from the real data. For example such artificial data could induce a bias in the RMSD minimization procedure of section 5.2. This is why we discarded data corresponding to layering artifacts without replacing them.

*9. Caption to Fig 5: You need to say explicitly WHICH signals you convoluted, e.g. green solid is probably the convolution of the WDC CH4 with the Dome C GAD estimated for LGM of Köhler et al 2011.*

We will modify the text to: 'WDC CH$_4$ signal convolved with different GADs: the Dome C GAD estimated for last deglaciation in green, etc'. We will also add the new GAD estimate by Köhler et al. (2015) in Fig. 6 and the result of its convolution with the WDC methane record in Fig. 5.

*10. Page 16, line 20: "modifying its two parameters", probably refers to the same 2 parameters given above in Eq 1. Please state, which values you choose in the end.*

Yes, the two parameters we optimize are the location and scale ( $\sigma$  and  $\mu$  in Eq. 1 provided by the Referee). The optimization is multivariate and finds the best (in the sense of RMSD minimization) location and scale simultaneously.
We will provide a Table with the parameters of all the used log-normal GADs (location, scale, mean and standard deviation)

*11. Fig 6: Needs a new GAD based on the Marcott WDC-EDC CO2 comparison, and/or the new approach of Köhler 2015.*

We will add the new GAD estimate by Köhler et al. (2015) in Figure 6 and the resulting smoothed signal in Figure 5.

*12. SI: Please either put all Figures to the end, or in the section, in which they are discussed.*

All figures will be placed at the end of the supplement.

*13. Please check references to Figures in main text, on SI page 5, line 4 a reference is given to Figre 6, but the correct Figure refered to here is Figure 5.*

We thank the referee for noticing this error. The references will be checked and made consistent.

---

## Author Comment (AC2) · 27 Sep 2017

**RESPONSE TO REFEREE 2:**

Please find below the reponses to the review of referee 2 on *Analytical constraints on layered gas trapping and smoothing of atmospheric variability in ice under low accumulation conditions.* The blue italic text is the text of the review, and the corresponding responses are below in black. When we intend to change the manuscript text or figures, it is stated so in the response.

*This study presents a high quality, novel data set consisting of ultra-high resolution methane measurements across Dansgaard-Oeschger event 17 in the low accumulation Vostok ice core from East Antarctica. The incredible detail of this record reveals rapid, anomalous signals that do not reflect past atmospheric changes, but are instead related to the process of time-varying gas trapping in the firn column. The authors develop a simple but effective numerical model to simulate the formation of these gas trapping artifacts, facilitating their removal and obtainment of a solely atmospheric signal. The Vostok atmospheric signal contains more high frequency information than would be expected from existing firn model-based predictions. A revised, much narrower, estimate of the gas age distribution at Vostok is produced. Although more work is needed to confirm these findings, the implication is that more detailed atmospheric records can be obtained from the older ice located in the Antarctic interior. The paper will be of interest to many in the ice core community and its implications are particularly relevant for the future development of CFA gas measurements and the search for the oldest ice. It is well written with excellent figures. I include many comments, but they should be straightforward for the authors to address.*

*Understanding gas trapping Can any more information be provided about the frequency of the gas trapping artifacts in depth and ice age domain? The signals reported by Rhodes et al. 2016 were annual – is the variability closer to decadal here and what does this suggest about the physical heterogeneity responsible? Is any comparison with high resolution chemistry possible across this interval?*

Contrary to Rhodes et al., 2016, a spectral analysis of the detrended noise (CFA data points minus spline values) did not show any spike around annual, decadal or any other time scale in our data. This will be mentioned in the manuscript. Nonetheless, as mentioned in the article p6 l28 of the manuscript, layering artifacts have a width roughly comparable to the annual accumulation. High resolution chemistry measurements are not available for the Vostok 4G2 ice core, and the ice dedicated to the project did not allow for additional analyses.

*Simple model of layered trapping Section 4.3. - Please provide more explanation of how extrapolation of Hörhold data to obtain density variability is carried out. What does the range in density variability obtained represent?*

We will add that 'Hörhold et al. (2011), propose linear regressions of the close-off density variability as a function of accumulation and temperature, based on various sites. Their lowest accumulation site is Dome C with an accumulation of 2.5 cm ice $yr^{-1}$ and a density variability of 4.6 kg $m^{-3}$. Applied to Vostok DO-17 conditions, the accumulation based extrapolation leads to a variability of 7 kg $m^{-3}$ and the temperature based extrapolation leads to a variability of 2.7 kg $m^{-3}$. This defines our extreme values (7 and 3 kg $m^{-3}$), and we chose the middle number of 5 kg $m^{-3}$ as the best value.'

*- Pg. 11, lines 20, 22-25: is the "closure depth shift" the difference in depth of pore closure between adjacent layers of different density? Please state clearly if so. And is the "age shift" of 207 yrs equivalent to the age difference between the gas trapped in adjacent layers? Does this value change as a result of subsequent tests? Pg. 13, line 4: are "age anomalies" the same as "age shift"?*

Age anomalies and age shifts are the same thing. The text and figure captions will be modified to use only the term 'depth anomaly' and 'age anomaly'.

The 'closure depth anomaly' is the difference in pore closure depth between an abnormal layer and an adjacent layer following the bulk behavior. Similarly, the 'age anomaly' of 207 years is the typical gas age difference between an abnormal layer and an adjacent layer following the bulk behavior. We will add these definitions to the manuscript.

This age anomaly is modified in the sensitivity tests and is indicated in the caption of corresponding supplementary figures.

*- A series of tests are conducted to illustrate the sensitivity of the model to input parameters. This is important and interesting but not that clear. Add some paragraphs please. Line 3 – make it clear that the extreme values used are the max and min of ranges already stated. A table including the parameters used and the resulting age and depth shifts would be informative.*

Our Section 4.3 discussing the layered trapping model will be clarified. The relationship between density variability, densification rate, depth anomaly and age anomaly values will be clarified. A table will be provided to summarize the parameters used in the sensitivity tests

*- Can anything be said about the relative importance of accumulation rate and density variability in causing gas trapping? Sites like Vostok have low accumulation, causing higher CH4 anomalies than high accumulation sites, but cold, low accumulation sites also tend to have lower density variability at depth (Fig. 7F, Horhold et al. 2012), which would cause lower CH4 anomalies.*

We will add to the text that 'under the hypothesis of density based layering, age anomalies strongly depend on accumulation as explained by Rhodes et al. (2016). A lower accumulation leads to a weaker density variability in the firn (Hörhold et al., 2011), but at the same time leads to a larger age difference between successive firn layers due to a steeper age-depth slope. The second effect tends to dominate and the net effect of a lower accumulation is an increase in age anomalies due to layered trapping.'

However, note that the relationship between age anomalies and concentration anomalies is dependent on the shape of the atmospheric signal.

*Estimation of Vostok GAD Section 5.2 - Needs an existing high(er) resolution CH4 record. No record exists beyond ∼100 ka (NEEM), which limits application of this technique. Abstract (line 14) should be modified to state need for higher resolution record. Still, it will be really interesting to see method applied to other sites for the Last Glacial.*

The abstract will be modified to specify that the method is based on the comparison with a weakly smoothed record.

*- Pg. 17, line 10-11. WD also experiences stable conditions over this time period. What if the reference atmospheric scenario was from NEEM where accumulation and temperature change greatly across DO events? Would method need to be adapted?*

The smoothing of ice core signal appears when the time scale of the atmospheric events is similar to the time scale of gas trapping in firn, or faster. Very short atmospheric events (e.g seasonal

variations) are never recorded in the ice as they are already smoothed out by diffusion in the open porosity of the firn (see e.g. Petrenko et al., 2013). Partial smoothing occurs near a cut-off frequency related to the duration of gas trapping, which is related to the temperature and accumulation rate. Our GAD estimation method works as long as the cut-off frequency of the high accumulation site is much higher that the cut-off frequency of the low accumulation record. In this case the highest frequencies of the high accumulation record are smoothed out at the low accumulation site regardless of their time variability. The major inconvenient of using NEEM would thus be the need to assess the inter-polar gradient in methane concentration.

*- Related to this, how valid is the assumption that WD represents the atmosphere? Why isn't this record also biased by gas trapping effects (high accumulation so faster trapping? more CFA smoothing?)?*

Rhodes et al. (2015), state that 'Only at gas ages > 60 ka BP is there a possibility that the continuous measurement system caused dampening of the $CH_4$ signal greater than that already imparted by firn-based smoothing processes'. Moreover, Figure S1 of their supplement predicts a GAD width of about 40yrs for the DO17 event, far beyond the width of the Vostok GAD. This ensures that the WD signal resolves enough high frequencies to be used as the weakly smoothed 'atmospheric' scenario for the Vostok ice core. This information will be added to the manuscript.

*- Pg. 19, line 2. Is the impact of layering on GAD really "unknown". Mitchell et al. (2015) state "total net effect of layering on gas trapping and the width of the age distribution of gases are unquestionably to narrow the age distribution" and your results seem to support this.*

We will replace the sentence 'Moreover, the impact of layering on the overall gas age distribution is unknown, and the Vostok record of DO-17 event strongly suggests an important layering effect even in very arid conditions.' p19 l3, with 'The weaker than expected smoothing during DO17 at Vostok could be due to the presence of a strong layering preventing air renewal and mixing, as suggested in Mitchell et al., (2015)'.

*Some discussion of the modelling work in Mitchell et al. (2015) might help the discussion here. Also on Pg. 21, line 18 – a sentence or two summarizing the findings of Mitchell et al. (2015) would help will the argument that firn models currently do a poor job (or don't attempt) at dealing with layered gas trapping.*

The sentence 'On the other hand, gas trapping processes are still weakly constrained in firn models (e.g. Mitchell et al., 2015)' will be replaced with 'On the other hand, Mitchell et al. (2015) point out the lack of firn layering representation in most firn models and conclude that firn layering narrows gas age distributions in ice.'

*Specific comments: Pg. 2, Lines 3-10: Consider stating that this is the 'traditional' description of the firn column. There is evidence, including the gas trapping anomalies presented here, that contradict the idea of bubble closure only occurring in the lock-in zone.*

The term traditional will be added: 'From a gas point of view, the firn is traditionally divided…'

*Pg. 3, line 27: Clathrate relaxation cavities are not mentioned again until Pg. 10. line 15. Sentence 'samples showed small clathrate relaxation cavities, the CFA sticks did not reveal visual anomalies'. Isn't this statement contradictory? In which direction would clathrate relaxation affect the CH4 signal and why?*

We will specify that by visual anomalies we meant visual stratification of the core, and local features. Clathrates were ubiquitous, and therefore not specific to anomalous layers.

If large enough, clathrate relaxation cavities could potentially lead to contamination by kerosene and/or outside air. However, the CFA setup only measures melt water from the inside of the ice core stick, in order to avoid this kind of contamination.

*Pg. 3, line 7: 'in periods of fast atmospheric variations...' Be clearer about what this means. Atmospheric variation must occur over the time frame of the gas trapping process (not seasonal variability for example), which will change with ice core analyzed.*

We will add: 'at a similar time scale as the gas trapping processes'.

*line 10: WAIS Divide information in Rhodes et al. (2016) is from a model only.*

We will remove the statement and rewrite the sentence to point out that modelling is used for WAIS Divide : ' Based on observations in high accumulation Greenland ice cores, and modeling for the WAIS Divide ice core, Rhodes et al. (2016) report that such artifacts can reach 40 ppbv in the methane ($CH_4$) record during the industrial time.'

*Pg. 9 & Fig. 3: Great figure. Could an arrow be added to indicate the direction of time (right to left)? In discussion about relative influence of early and late closure on final signal, do you mean proportion of early trapped layers will be greater than later trapped layers? Or, do you mean the amplitude of the early layer signals will be greater than the later layer signals? Could this hypothesis be illustrated on the figure?*

We expect the proportion of late and early pore closure to be the same. However, a late pore closure means that the surrounding firn is sealed and prevents long distance gas transport. The latest closure layers will not be able to trap young air if gas transport is impossible in the surrounding firn layers, resulting in less important artifacts.

The text will be modified to better explain this point.

We have thought about displaying asymmetrical artifacts on the Figure 3 to illustrate this effect, but it does not explain the underlying mechanism. An arrow will be added on the Figure, to indicate the direction of time.

*Pg. 12 & Pg. 16: Buizert et al. (2012) does not convert gas ages to AICC2012. Do you mean GICC05 here?*

More details on gas age scale conversions will be provided in answer to comments by both referees (see answer to minor comment number 6 by Referee#1).

*Figure 5 & Pg. 15: Yes, the tiny sub-centennial variation is smoothed out in Vostok, but multi-centennial information is preserved, e.g., feature 58.7-58.4 ka. This is more detail than we would expect from Dome C GAD estimation and worth mentioning. It would help justify statement on Pg. 19, line 9 that at the moment is tenuous.*

We will comment on the multi-centenial features preserved.

*Pg. 20, line 1: Can you be more specific about the 'bias' possibly introduced by gas trapping*

*artifacts? Does the 7 ppbv refer to a positive or negative bias? Wouldn't the direction of bias change with the atmospheric trend and so even itself out over the relatively short timescales of gas trapping (compared to length of record compressed within small depth of ice)?*

To be clearer we will change the text to 'In the very simple case of a record with artifacts all negatively orientated, covering 15% of the ice core and all reaching 50 ppbv, this bias is about -7 ppbv.'

If artifacts are distributed on each side of the record they will then partially even out, but a biais might exist nonetheless.

*Supplement, Pg. 1, line 13: statement about WAIS data being scaled to discrete measurements is not accurate.*

There was indeed an error here. Rhodes et al, 2015 calibrated their measurements for methane dissolution, but not by using discrete measurements. The text will be changed.

*Technical notes: Pg. 1, Line 13: Add "numerical" method.*

It will be added.

*Pg. 1, Line 21 and repeatedly through manuscript: "gases get enclosed within bubbles...and allow reconstructing..." "Allow reconstructing" is not grammatically correct and should be changed to something like "allow us to reconstruct..." or "allowing reconstruction of...".*

The text will be modified accordingly.

*Pg. 2, line 11: change "atmospheric composition events" to "atmospheric variability"*

The text will be modified accordingly.

*Pg. 2 line 12: "dampening" should read "damping" = the decrease in the amplitude of an oscillation or wave motion with time.*

The text will be modified accordingly.

*Pg. 3, line 4: define or explain "short scale physical variability"*

Short scale physical variability refers to centimeter scale variability. This will be added to the text.

*Pg. 3, line 5: insert "physical" before heterogeneities*

The text will be modified accordingly.

*Pg. 4, line 2-3 repeats what is said on previous page*

The sentence 'It was selected to include the Dansgaard-Oeschger event 17, showing a rapid and large increase in atmospheric methane concentration of about 150 ppbv within 500 yr (Brook et al., 1996; Chappellaz et al., 2013; Rhodes et al., 2015).' will be removed from the manuscript.

*Pg. 4, line 12: state volume of debubbler*

The debubbler we used is a T-shaped manifold and do not have a headspace for gas to mix in. Hence, the volume does not appear to us as an important parameter. The manuscript will be modified to include the shape of our debubbler.

*Pg. 6, line 5 onwards: separate into two paragraphs*

The text will be modified accordingly.

*Pg. 6, line 15: does 2.1m represent one instance of kerosene contamination or it is the sum of many?*

In total, 2.1m of data were lost due to kerosene contamination. The text will be modified to 'Adding the length of all kerosene affected ice core sections, a total of 2.1 m of data was removed.'

*Fig. S3: Add indication of depth range represented.*

We will add in the caption that the length of ice melted in the data shown is about 25cm.

*Fig. 1: increase sub-figure size*

It will be done.

*Fig. S11 caption and elsewhere: be specific - 'WD2014' gas chronology*

The name of the chronology will be changed accordingly.

*Pg. 6, Line 21: change to "atmosphere relevant" of atmospheric history relevant?*

It will be done.

*Pg. 6, line 28: 50 ppbv amplitude and 2 cm wavelength*

We will add the word amplitude. However, artifacts are not sinusoidal or periodic, and thus we prefer to keep using the term width rather than wavelength.

*Pg. 8, line 26: 'the closure of such a layer is likely progressive' – please clarify meaning*

The text will be modified to be clearer.

*Pg. 9, line 4: clarify 'significant atmospheric variations'. Again, quantify.*

We will remove the word significant and change to 'during periods of variations in concentrations of atmospheric gases'.

*Pg. 11, line 9: replace 'later' with 'latter'*

It will be done.

*Pg. 13, line 21: a signal that is representative of only atmospheric variability*

The text will be modified to 'To extract an undisturbed (chronologically monotonous and representative of atmospheric variability only) [...]'.

*Pg. 13, line 28: be specific here, artifacts already removed are due to breaks or kerosene.*

'Already removed artifacts' refers to layering artifacts. The code does not necessarily clean all layering artifacts at the first iteration of the looping algorithm, and might require to further treat an already partially cleaned signal. The looping procedure will be better introduced and the text will be modified to '(with or without already partially removed layering artifacts during the cleaning process)'.

*Pg. 14, line 3: provide details on spline fit*

In relation with this comment and minor comment No7 of Referee#1, the text will be changed to: 'A spline of degree 3 is used to interpolate between the binned points on the original CFA depth scale. This interpolating spline does not further smooth the signal, and is used as a guess of the chronological signal.'

*Pg. 14, line 27: 'high frequency atmospheric variability'*

It will be done.

*Pg. 19, line 23: be specific here, "anomalous layers" are 'gas trapping artifacts' or artifacts also due to other things like kerosene?*

We meant 'anomalous layers' to refer to layers with gas trapping artifacts. The text will be changed to 'However, continuous flow analysis may not always allow us to distinguish between layering artifacts and the chronologically ordered signal.'

*Pg. 21, line 21: delete "or infirm" Strictly, the WAIS Divide ice core should be referred to as WD, not WAIS (the ice sheet).*

'Or infirm' will be removed. The WAIS Divide core will be referred as WDC or WD ice core in the text.

Reference:

Petrenko V. V., Martinerie P., Novelli P., Etheridge D. M., Levin I., Wang Z., Blunier T. et al. "A 60 yr record of atmospheric carbon monoxide reconstructed from Greenland firn air." *Atmospheric Chemistry and Physics* 13, no. 15 (2013): 7567-7585.

---

## Author Response (AR1)

Dear Pr. Fischer,

Please find enclosed a revised version of manuscript cp-2017-78 entitled: *Analytical constraints on layered gas trapping and smoothing of atmospheric variability in ice under low accumulation conditions.*
The responses to the reviews of referees 1 and 2 copied below are the same as those published on the Clim. Past Discuss web page, with added references to the pages and lines of the revised manuscript where the changes appear. The blue italic text is the text of the reviews, and the corresponding responses are below in black.

In the revised manuscript, the text removed appears as red strikethrough, and the text added appears in blue.

Please note that we modified the bibliography at the end of the article as well. The articles cited are the same (except for the added citation of Köhler et al., 2015), but we corrected a few mistakes and made the style of journal names and doi links more consistent with the Clim. Past display.
We also corrected some typographic errors found in the manuscript.

Sincerly,
Kévin Fourteau on behalf of all co-authors.

**RESPONSE TO REFEREE 1:**

*This paper analyses CH4 across Dansgaard/Oeschger event 17 (DO-17) from the Vostok ice core with a CFA-based measurement system in order to improve understanding of layered gas trapping and smoothing of atmospheric variability in an ice core drilled in low accumulation areas. A thus dervied CH4 record is then postprocessed and finally compared with CH4 from the higher resolving WAIS Divide ice core (WDC) to conclude that gas age distribution (GAD) - or smoothing - in Vostok seems to be similar for modern and DO-17 conditions. The paper is well written, especially the post-processing procedure described to some detail. The detection of artifacts in CH4 from such a high-resolution system as presented here is convincing.*

*However, the paper falls so far short in one aspect, that is the application of a previously assumed LGM gas age distribution used for EPICA Dome C (EDC) to transfer WDC CH4 data into potentially signals recorded in Vostok, from which it was concluded, that gas age distribution are probably independent from climate background.*

Our conclusion that smoothing during DO17 is weaker than expected is mostly based on the comparison between the gas age distribution (GAD) tuned to our $CH_4$ data and the GAD estimated by Witrant et al., 2012 for modern conditions at Vostok.

*Here, they use what has been used as gas age distribution in Köhler et al (2011), who used a log-normal function, and for LGM assumed a mean width of the GAD of 590 years. This GAD allowed large overshoots in the true atmospheric signal of CO2, when compared with the EDC ice core record of CO2, and was again used in Köhler et al. (2014). However, the new WDC CO2 paper of Marcott et al (2014) showed, that the assumed GAD used by Köhler in 2011 was probably too wide since a much smaller GAD was able to transfer the WDC CO2 (potentially very close to the true atmospheric variability of CO2) to the CO2 record obtained from EDC (Extended Data Fig 5 in Marcott et al., 2014). This revised narrow GAD was also then applied for the question of interest by*

*Köhler, but I have to admit, so far only published in a conference proceeding, not widely known (Köhler et al., 2015, pages 135–140 in http://www.leopoldina.org/uploads/tx_leopublication/NAL_Bd121_Nr408_LR.pdf). Figure 2 of this 6-pages proceeding contains a transformation of a simulated atmospheric CO2 into a signal recorded at EDC around the onset of the Bølling/Allerød (B/A) warm period around 14.6 kyr BP using the same log-normal function as introduced in Köhler et al (2011) of*

$$y = \frac{1}{x \cdot \sigma \cdot \sqrt{2\pi}} \cdot e^{-0.5(\frac{ln(x)-\mu}{\sigma})^2} \qquad (1)$$

*with x (yr) as the time elapsed since the last exchange with the atmosphere, which leads to an expected value (mean) E of the GAD of*

$$E = e^{\mu + \frac{\sigma}{2}}. \qquad (2)$$

*From the two free parameters μ and σ, in 2011 Köhler has chosen for simplicity σ = 1, but now in the revised application in 2015 uses σ < 1 to reproduce the shape of the GAD suggested in Marcott. In detail σ = 0.5 was used and μ defined in a way which guarantees the pre-defined mean values E of 150 yr. So, not only the mean width of the GAD has be reduced by a factor of 2.7, from formally 400 yrs to now 150 yrs (for this application to the B/A), but also the shape of the GAD. I believe the authors are challenged now to also use a GAD that agrees with the WDC-EDC CO2 comparsion, as brought up by Marcott, and in a first step probably at best also start with a revised log-normal function using different parameter values. Only then can they conclude (or not) if the GAD is indeed similar for modern and DO-17 climate or not. For any such exercise, please always state the used parameter values of the function, e.g. as given here, both the chosen form-shape factor σ, and at best the mean value E (directly derived from μ once σ is given). So far, no details on the applied log-normal function has been given. This will probably lead to a revision of the final conclusion and figure 5, but the rest of the paper is largely unaffected.*

We added the new GAD by Köhler et al., 2015 in Figures 5 and 6 (p17 and 19 of the revised manuscript). This new GAD leads to a somewhat lower smoothing than the modern one based on Witrant et al. (2012). But there are large uncertainties in GAD estimation as we mention p20 l16-19 of the revised manuscript. However, the main conclusion is unaffected. The smoothing during DO17 is comparable to the modern one at Vostok, despite the lower accumulation.
We modified the discussion of smoothing between the various available GADs to include the Köhler et al., 2015 GAD, and stated more clearly that the relation between accumulation and smoothing does not appear monotonous (p20 l5- 13 of the revised manuscript).
We will also add a Table with the parameters of all the GADs used to facilitate the comparison with other estimates (p20 of the revised manuscript). Note that we did not assume $\sigma=1$ but adjusted this parameter in our estimates.

*Further minor comments in chronological order:*

*1. Throughout the paper: Units are sometimes weight, with a dot (.) inbetween, e.g. "3.8 cm.min−1", which should be "3.8 cm min−1 ".*

We removed the dot in between units.

*2. Figure 1: Labels in insert (top right corner) are much too small.*

Figure 1 was redrawn with larger zoom and labels. (p7 of the revised manuscript)

*3. Page 9, line 4: " As explained in Rhodes et al. (2016), such a mechanism affects trace gases record only during periods of significant atmospheric variations." Variations of what? Probably "variations in concentrations of atmospheric gases".*

Yes we meant variations in concentrations of atmospheric gases. The text was modified accordingly to 'during periods of variations in concentration of atmospheric gases'. (p9 l10 of the revised manuscript)

*4. Page 9, line 11: "monotonous variations", change to "monotonous in/decrease".*

The text was changed to 'monotonous increase/decrease'. (p10 l4 of the revised manuscript)

*5. Page 11, line 21: What are the coldest sites in Breant et al (2016)? Please name here.*

The sites are Dome C, Vostok, and Dome A. This was added to the text. (p12 l10 of the revised manuscript)

*6. Page 12, line 4: "... the methane record from the WAIS Divide ice core (Rhodes et al., 2015), with gas ages with gas ages converted on the AICC2012 scale (Buizert et al., 2015)." Now, this needs some more explanation and probably correction. Buizert et al., 2015 does NOT plot WDC $CH_4$ on AICC2012, as suggested by this sentence. There is also the effort of explaining the gas age adaption of WDC $CH_4$ to AICC2012 in the SI Fig S11 (and corresponding SI section), which I also did not understand in detail. Please be precise here, and describe this step in the main text, not hidden in the SI.*

We added to the manuscript: 'The WDC gas age chronology (WD2014) was scaled to the GICC05 chronology (with present defined as 1950) dividing by a factor of 1.0063 as in Buizert et al. (2015). For the rest of the article we used this scaled WD2014 chronology to express WDC gas ages.' (p12 l22-24 of the revised manuscript)

*7. Page 14, line 3: I believe, a spline normally comes along with a cutoff-frequency, which has not been given here.*

We are using an interpolating spline, which does not smooth the signal but goes through each data point. The smoothing is taken care of by the averaging over 50cm wide bins, and this averaging length works as the cut-off frequency. The text was changed to 'A spline of degree 3 is used to interpolate between the binned points on the original CFA depth scale. This interpolating spline does not further smooth the signal, and is used as a guess of the chronological signal.' (p15 l12-13 of the revised manuscript)

*8. Section 4.4 (removing artifacts) page 13-14, versus Fig 1. My understanding of the description of Section 4.4 was, that the spikes caused by layering artifacts are removed, and a continous $CH_4$ time series without artifacts is generated. However, the black line in Fig 1 (which according to the text should be such a time series) does not contain any data in the periods, in which artifacts has been removed. I would think the post-processing should give you some data points in exchange to the removed artifical peaks. Please refine text, or change Figure 1 accordingly.*

Even in the absence of layering artifacts, the CFA signal is discontinuous due to missing ice sections, kerosene and ambient air infiltrations. This fact was specified in the text. (p6 l19-20 of the

revised manuscript)

During cleaning of layering artifacts, we do not wish to mix the experimental data with interpolated data that are not independent from the real data. For example such artificial data could induce a bias in the RMSD minimization procedure of section 5.2. This is why we discarded data corresponding to layering artifacts without replacing them.

*9. Caption to Fig 5: You need to say explicitly WHICH signals you convoluted, e.g. green solid is probably the convolution of the WDC CH4 with the Dome C GAD estimated for LGM of Köhler et al 2011.*

We modified the text of the caption to: 'WDC CH$_4$ signal convolved with different GADs: the Dome C GAD estimated for the Bølling-Allerød by Köhler et al. (2015) in red, etc'. (p17 of the revised manuscript)

We also added the new GAD estimate by Köhler et al. (2015) in Fig. 6 and the result of its convolution with the WDC methane record in Fig. 5. (p17 and 19 of the revised manuscript)

*10. Page 16, line 20: "modifying its two parameters", probably refers to the same 2 parameters given above in Eq 1. Please state, which values you choose in the end.*

Yes, the two parameters we optimize are the location and scale ( $\sigma$ and $\mu$ in Eq. 1 provided by the Referee). The optimization is multivariate and finds the best (in the sense of RMSD minimization) location and scale simultaneously.

We provided a Table with the parameters of all the used log-normal GADs (location, scale, mean and standard deviation). (p20 of the revised manuscript)

*11. Fig 6: Needs a new GAD based on the Marcott WDC-EDC CO2 comparison, and/or the new approach of Köhler 2015.*

We added the new GAD estimate by Köhler et al. (2015) in Figure 6 and the resulting smoothed signal in Figure 5. (p17 and 19 of the revised manuscript)

*12. SI: Please either put all Figures to the end, or in the section, in which they are discussed.*

All figures were placed at the end of the supplement.

*13. Please check references to Figures in main text, on SI page 5, line 4 a reference is given to Figre 6, but the correct Figure refered to here is Figure 5.*

We thank the referee for noticing this error. The references were checked and made consistent.

**RESPONSE TO REFEREE 2:**

*This study presents a high quality, novel data set consisting of ultra-high resolution methane measurements across Dansgaard-Oeschger event 17 in the low accumulation Vostok ice core from East Antarctica. The incredible detail of this record reveals rapid, anomalous signals that do not reflect past atmospheric changes, but are instead related to the process of time-varying gas trapping in the firn column. The authors develop a simple but effective numerical model to simulate the formation of these gas trapping artifacts, facilitating their removal and obtainment of a solely atmospheric signal. The Vostok atmospheric signal contains more high frequency information than would be expected from existing firn model-based predictions. A revised, much narrower, estimate of the gas age distribution at Vostok is produced. Although more work  is needed to confirm these*

*findings, the implication is that more detailed atmospheric records can be obtained from the older ice located in the Antarctic interior. The paper will be of interest to many in the ice core community and its implications are particularly relevant for the future development of CFA gas measurements and the search for the oldest ice. It is well written with excellent figures. I include many comments, but they should be straightforward for the authors to address.*

*Understanding gas trapping Can any more information be provided about the frequency of the gas trapping artifacts in depth and ice age domain? The signals reported by Rhodes et al. 2016 were annual – is the variability closer to decadal here and what does this suggest about the physical heterogeneity responsible? Is any comparison with high resolution chemistry possible across this interval?*

Contrary to Rhodes et al., 2016, a spectral analysis of the detrended noise (CFA data points minus spline values) did not show any spike around annual, decadal or any other time scale in our data. This was added in the manuscript. (p11 l14-15 of the revised manuscript)
Nonetheless, as mentioned in the revised article p7 l3, layering artifacts have a width roughly comparable to the annual accumulation.
High resolution chemistry measurements are not available for the Vostok 4G2 ice core, and the ice dedicated to the project did not allow for additional analyses.

*Simple model of layered trapping Section 4.3. - Please provide more explanation of how extrapolation of Hörhold data to obtain density variability is carried out. What does the range in density variability obtained represent?*

We added that 'Hörhold et al. (2011), propose linear regressions of the close-off density variability as a function of accumulation and temperature, based on various sites. Their lowest accumulation site is Dome C with an accumulation of 2.5 cm ice $yr^{-1}$ and a density variability of 4.6 kg $m^{-3}$. Applied to Vostok DO-17 conditions, the accumulation based extrapolation leads to a variability of 7 kg $m^{-3}$ and the temperature based extrapolation leads to a variability of 2.7 kg $m^{-3}$. This defines our extreme values (7 and 3 kg $m^{-3}$), and we chose the middle number of 5 kg $m^{-3}$ as the best value.' (p12 l1-6 of the revised manuscript)

*- Pg. 11, lines 20, 22-25: is the "closure depth shift" the difference in depth of pore closure between adjacent layers of different density? Please state clearly if so. And is the "age shift" of 207 yrs equivalent to the age difference between the gas trapped in adjacent layers? Does this value change as a result of subsequent tests? Pg. 13, line 4: are "age anomalies" the same as "age shift"?*

Age anomalies and age shifts are the same thing. The text and figure captions were modified to use only the term 'depth anomaly' and 'age anomaly'.

The 'closure depth anomaly' is the difference in pore closure depth between an abnormal layer and an adjacent layer following the bulk behavior. Similarly, the 'age anomaly' of 207 years is the typical gas age difference between an abnormal layer and an adjacent layer following the bulk behavior. We added these definitions to the manuscript. (p12 l8 and l14 of the revised manuscript)

This age anomaly is modified in the sensitivity tests and is indicated in the caption of corresponding supplementary figures.

*- A series of tests are conducted to illustrate the sensitivity of the model to input parameters. This is important and interesting but not that clear. Add some paragraphs please. Line 3 – make it clear that the extreme values used are the max and min of ranges already stated. A table including the*

*parameters used and the resulting age and depth shifts would be informative.*

Our Section 4.3 discussing the layered trapping model was clarified. We provided definitions for depth and age anomalies (p12 l8 and l14 of the revised manuscript), and explained how depth anomalies are computed using density variability and density depth gradient (p12 l12-13 of the revised manuscript).
A table was provided to summarize the parameters used in the sensitivity tests, their impacts on depth and age anomalies, and the associated figures (p14 of the revised manuscript).

*- Can anything be said about the relative importance of accumulation rate and density variability in causing gas trapping? Sites like Vostok have low accumulation, causing higher CH4 anomalies than high accumulation sites, but cold, low accumulation sites also tend to have lower density variability at depth (Fig. 7F, Horhold et al. 2012), which would cause lower CH4 anomalies.*

We added to the text that 'under the hypothesis of density based layering, age anomalies strongly depend on accumulation as explained by Rhodes et al. (2016). A lower accumulation leads to a weaker density variability in the firn (Hörhold et al., 2011), but at the same time leads to a larger age difference between successive firn layers due to a steeper age-depth slope. The second effect tends to dominate and the net effect of a lower accumulation is an increase in age anomalies due to layered trapping.' (p14 l12-15 of the revised manuscript)
We also removed the sentence p13 l16-17 of the revised manuscript mentioning the effect of accumulation, since a more detailed paragraph was included.

However, note that the relationship between age anomalies and concentration anomalies is dependent on the shape of the atmospheric signal.

*Estimation of Vostok GAD Section 5.2 - Needs an existing high(er) resolution CH4 record. No record exists beyond ∼100 ka (NEEM), which limits application of this technique. Abstract (line 14) should be modified to state need for higher resolution record. Still, it will be really interesting to see method applied to other sites for the Last Glacial.*

The abstract was modified to specify that the method is based on the comparison with a weakly smoothed record. (p1 l14 of the revised manuscript)

*- Pg. 17, line 10-11. WD also experiences stable conditions over this time period. What if the reference atmospheric scenario was from NEEM where accumulation and temperature change greatly across DO events? Would method need to be adapted?*

The smoothing of ice core signal appears when the time scale of the atmospheric events is similar to the time scale of gas trapping in firn, or faster. Very short atmospheric events (e.g seasonal variations) are never recorded in the ice as they are already smoothed out by diffusion in the open porosity of the firn (see e.g. Petrenko et al., 2013). Partial smoothing occurs near a cut-off frequency related to the duration of gas trapping, which is related to the temperature and accumulation rate. Our GAD estimation method works as long as the cut-off frequency of the high accumulation site is much higher that the cut-off frequency of the low accumulation record. In this case the highest frequencies of the high accumulation record are smoothed out at the low accumulation site regardless of their time variability. The major inconvenient of using NEEM would thus be the need to assess the inter-polar gradient in methane concentration.

*- Related to this, how valid is the assumption that WD represents the atmosphere? Why isn't this*

*record also biased by gas trapping effects (high accumulation so faster trapping? more CFA smoothing?)?*

Rhodes et al. (2015), state that 'Only at gas ages > 60 ka BP is there a possibility that the continuous measurement system caused dampening of the $CH_4$ signal greater than that already imparted by firn-based smoothing processes'. Moreover, Figure S1 of their supplement predicts a GAD width of about 40yrs for the DO17 event, far beyond the width of the Vostok GAD. This ensures that the WD signal resolves enough high frequencies to be used as the weakly smoothed 'atmospheric' scenario for the Vostok ice core. This information was added to the manuscript (p18 l12-16 of the revised manuscript).

*- Pg. 19, line 2. Is the impact of layering on GAD really "unknown". Mitchell et al. (2015) state "total net effect of layering on gas trapping and the width of the age distribution of gases are unquestionably to narrow the age distribution" and your results seem to support this.*

We replaced the sentence 'Moreover, the impact of layering on the overall gas age distribution is unknown, and the Vostok record of DO-17 event strongly suggests an important layering effect even in very arid conditions.', with 'The weaker than expected smoothing during DO17 at Vostok could be due to the presence of a strong layering preventing air renewal and mixing, as suggested in Mitchell et al., (2015)' (p21 l6-9 of the revised manuscript).

*Some discussion of the modelling work in Mitchell et al. (2015) might help the discussion here. Also on Pg. 21, line 18 – a sentence or two summarizing the findings of Mitchell et al. (2015) would help will the argument that firn models currently do a poor job (or don't attempt) at dealing with layered gas trapping.*

The sentence 'On the other hand, gas trapping processes are still weakly constrained in firn models (e.g. Mitchell et al., 2015)' was replaced with 'On the other hand, Mitchell et al. (2015) point out the lack of firn layering representation in most firn models and conclude that firn layering narrows gas age distributions in ice.' (p24 l1-2 of the revised manuscript)

*Specific comments: Pg. 2, Lines 3-10: Consider stating that this is the 'traditional' description of the firn column. There is evidence, including the gas trapping anomalies presented here, that contradict the idea of bubble closure only occurring in the lock-in zone.*

The term traditional was added: 'From a gas point of view, the firn is traditionally divided…' (p2 l6 of the revised manuscript).

*Pg. 3, line 27: Clathrate relaxation cavities are not mentioned again until Pg. 10. line 15. Sentence 'samples showed small clathrate relaxation cavities, the CFA sticks did not reveal visual anomalies'. Isn't this statement contradictory? In which direction would clathrate relaxation affect the CH4 signal and why?*

We specified that by visual anomalies we meant visual stratification of the core, and local features. Clathrates were ubiquitous, and therefore not specific to anomalous layers (p11 l9-10 of the revised manuscript).
If large enough, clathrate relaxation cavities could potentially lead to contamination by kerosene and/or outside air. However, the CFA setup only measures melt water from the inside of the ice core stick, in order to avoid this kind of contamination.

*Pg. 3, line 7: 'in periods of fast atmospheric variations...' Be clearer about what this means. Atmospheric variation must occur over the time frame of the gas trapping process (not seasonal variability for example), which will change with ice core analyzed.*

We added 'at a similar time scale as the gas trapping processes' (p3 l8 of the revised manuscript).

*line 10: WAIS Divide information in Rhodes et al. (2016) is from a model only.*

We removed the statement and rewrote the sentence to point out that modelling is used for WAIS Divide: ' Based on observations in high accumulation Greenland ice cores, and modeling for the WAIS Divide ice core, Rhodes et al. (2016) report that such artifacts can reach 40 ppbv in the methane ($CH_4$) record during the industrial time' (p3 l9-10 of the revised manuscript).

*Pg. 9 & Fig. 3: Great figure. Could an arrow be added to indicate the direction of time (right to left)? In discussion about relative influence of early and late closure on final signal, do you mean proportion of early trapped layers will be greater than later trapped layers? Or, do you mean the amplitude of the early layer signals will be greater than the later layer signals? Could this hypothesis be illustrated on the figure?*

We expect the proportion of late and early pore closure to be the same. However, a late pore closure means that the surrounding firn is sealed and prevents long distance gas transport. The latest closure layers will not be able to trap young air if gas transport is impossible in the surrounding firn layers, resulting in less important artifacts.
The text was modified to better explain this point (p10 l8-10 of the revised manuscript).

We have thought about displaying asymmetrical artifacts on the Figure 3 to illustrate this effect, but it does not explain the underlying mechanism. An arrow was added on the Figure, to indicate the direction of time (p10 of the revised manuscript).

*Pg. 12 & Pg. 16: Buizert et al. (2012) does not convert gas ages to AICC2012. Do you mean GICC05 here?*

More details on gas age scale conversions were provided in answer to comments by both referees (see answer to minor comment number 6 by Referee#1 and modification of manuscript p12 l22-24).

*Figure 5 & Pg. 15: Yes, the tiny sub-centennial variation is smoothed out in Vostok, but multi-centennial information is preserved, e.g., feature 58.7-58.4 ka. This is more detail than we would expect from Dome C GAD estimation and worth mentioning. It would help justify statement on Pg. 19, line 9 that at the moment is tenuous.*

We added a sentence on the multi-centennial features preserved (p16 l13-14 of the revised manuscript).

*Pg. 20, line 1: Can you be more specific about the 'bias' possibly introduced by gas trapping artifacts? Does the 7 ppbv refer to a positive or negative bias? Wouldn't the direction of bias change with the atmospheric trend and so even itself out over the relatively short timescales of gas trapping (compared to length of record compressed within small depth of ice)?*

To be clearer we changed the text to 'In the very simple case of a record with artifacts all negatively orientated, covering 15% of the ice core and all reaching 50 ppbv, this bias is about -7 ppbv.' (p22

l7-9 of the revised manuscript)

If artifacts are distributed on each side of the record they will then partially even out, but a bias might exist nonetheless.

*Supplement, Pg. 1, line 13: statement about WAIS data being scaled to discrete measurements is not accurate.*

There was indeed an error here. Rhodes et al, 2015 calibrated their measurements for methane dissolution, but not by using discrete measurements. The text was changed (p1 l13-14 of the revised supplement).

*Technical notes: Pg. 1, Line 13: Add "numerical" method.*

It was added (p1 l13 of the revised manuscript).

*Pg. 1, Line 21 and repeatedly through manuscript: "gases get enclosed within bubbles...and allow reconstructing..." "Allow reconstructing" is not grammatically correct and should be changed to something like "allow us to reconstruct..." or "allowing reconstruction of...".*

The text was modified accordingly.

*Pg. 2, line 11: change "atmospheric composition events" to "atmospheric variability"*

The text was modified accordingly (p2 l12 of the revised manuscript).

*Pg. 2 line 12: "dampening" should read "damping" = the decrease in the amplitude of an oscillation or wave motion with time.*

The text was modified accordingly throughout the manuscript.

*Pg. 3, line 4: define or explain "short scale physical variability"*

Short scale physical variability refers to centimeter scale variability. This was added to the text (p3 l4 of the revised manuscript).

*Pg. 3, line 5: insert "physical" before heterogeneities*

The text was modified accordingly (p3 l5 of the revised manuscript).

*Pg. 4, line 2-3 repeats what is said on previous page*

The sentence 'It was selected to include the Dansgaard-Oeschger event 17, showing a rapid and large increase in atmospheric methane concentration of about 150 ppbv within 500 yr (Brook et al., 1996; Chappellaz et al., 2013; Rhodes et al., 2015).' was removed from the manuscript (p4 l3-5 of the revised manuscript).

*Pg. 4, line 12: state volume of debubbler*

The debubbler we used is a T-shaped manifold and do not have a headspace for gas to mix in. Hence, the volume does not appear to us as an important parameter. The manuscript was modified to include the shape of our debubbler (p4 l15 of the revised manuscript).

*Pg. 6, line 5 onwards: separate into two paragraphs*

The text was modified accordingly (p6 l5 of the revised manuscript).

*Pg. 6, line 15: does 2.1m represent one instance of kerosene contamination or it is the sum of many?*

In total, 2.1m of data were lost due to kerosene contamination. The text was modified to 'Adding the length of all kerosene affected ice core sections, a total of 2.1 m of data was removed.' (p6 l16-17 of the revised manuscript)

*Fig. S3: Add indication of depth range represented.*

We added in the caption that the length of ice melted in the data shown is about 25cm (p5 of the revised supplement).

We also added the length of the thin sections in the caption of Figure S5 (p5 of the revised supplement).

*Fig. 1: increase sub-figure size*

Figure 1 was redrawn with a larger zoom (p7 of the revised manuscript).

*Fig. S11 caption and elsewhere: be specific - 'WD2014' gas chronology*

The name of the chronology was changed accordingly.

*Pg. 6, Line 21: change to "atmosphere relevant" of atmospheric history relevant?*

It was be done (p6 l23 of the revised manuscript).

*Pg. 6, line 28: 50 ppbv amplitude and 2 cm wavelength*

We added the word amplitude (p7 l3 of the revised manuscript). However, artifacts are not sinusoidal or periodic, and thus we prefer to keep using the term width rather than wavelength.

*Pg. 8, line 26: 'the closure of such a layer is likely progressive' – please clarify meaning*

The text was modified to be clearer (p9 l4 of the revised manuscript).

*Pg. 9, line 4: clarify 'significant atmospheric variations'. Again, quantify.*

We will remove the word significant and change to 'during periods of variations in concentrations of atmospheric gases' (p9 l10 of the revised manuscript).

*Pg. 11, line 9: replace 'later' with 'latter'*

It was be done (p11 l20 of the revised manuscript).

*Pg. 13, line 21: a signal that is representative of only atmospheric variability*

The text was modified to 'To extract an undisturbed (chronologically monotonous and representative of atmospheric variability only) […]' (p15 l1 of the revised manuscript).

*Pg. 13, line 28: be specific here, artifacts already removed are due to breaks or kerosene.*

'Already removed artifacts' refers to layering artifacts. The code does not necessarily clean all layering artifacts at the first iteration of the looping algorithm, and might require to further treat an already partially cleaned signal. The looping procedure was better introduced and the text was modified to '(with or without already partially removed layering artifacts during the cleaning process)' (p15 l7 and l9-10 of the revised manuscript).

*Pg. 14, line 3: provide details on spline fit*

In relation with this comment and minor comment No7 of Referee#1, the text was changed to: 'A spline of degree 3 is used to interpolate between the binned points on the original CFA depth scale. This interpolating spline does not further smooth the signal, and is used as a guess of the chronological signal.' (p15 l12-13 of the revised manuscript).

*Pg. 14, line 27: 'high frequency atmospheric variability'*

It was done (p16 l8 of the revised manuscript).

*Pg. 19, line 23: be specific here, "anomalous layers" are 'gas trapping artifacts' or artifacts also due to other things like kerosene?*

We meant 'anomalous layers' to refer to layers with gas trapping artifacts. The text was changed to 'However, continuous flow analysis may not always allow us to distinguish between layering artifacts and the chronologically ordered signal.' (p22 l1 of the revised manuscript)

*Pg. 21, line 21: delete "or infirm" Strictly, the WAIS Divide ice core should be referred to as WD, not WAIS (the ice sheet).*

'Or infirm' was removed (p24 l5 of the revised manuscript).
The text was modified to refer to the WAIS Divide core as WDC.

Reference:

Petrenko V. V., Martinerie P., Novelli P., Etheridge D. M., Levin I., Wang Z., Blunier T. et al. "A 60 yr record of atmospheric carbon monoxide reconstructed from Greenland firn air." *Atmospheric Chemistry and Physics* 13, no. 15 (2013): 7567-7585.

---

## Author Response (AR2)

Dear Professor Fischer,

Please find the detailed responses to your comments in the notice and in the modified manuscript below, with modifications highlighted in the text. Green highlighted text is used when we directly accepted your proposed correction. Yellow highlighted text is used when we modified your proposed correction, or when there was no proposed correction. Finally removed text without replacement is displayed as red strikethrough. Note that the latex text highlighting might results in some problems in the page layeout. They are however not present in the proposed final version.

On behalf of all the authors,
Kévin Fourteau.

*- p11 l14: in contrast to the reviewer I think this is not important and comes out of the blue here, potentially delete*

We will remove this sentence.

*- p12 l3: Is there an error here. Hoerhold predicts lower variability for lower acc, but your Vostok variability estimate is now larger than at Dome C despite lower acc???*

Hörhold proposed linear laws which indeed result in lower variabilities for lower accumulations (and temperatures). The values we used are based those laws. However the experimental points are significantly scattered around the linear regressions (R = 0.738 and 0.822 for the accumulation and temperature based laws respectively). The experimental point corresponding to Dome C has a particularly low value and falls below the regression lines, as displayed in Figure 9 of Hörhold et al, 2011. This is why we use a large range of density variabilities for the sensitivity tests.

*- p12 l15:I can't follow the argumentation here. Please clarify*

As explained as the end to the section 4.1, late pore closure anomalies tend to produce weaker artefacts than early ones, as the surrounding firn is already sealed and prevents air renewal and mixing.
As our model does not explicitly represent gas transport, we represent this sealing effect by applying a factor of 0.25 to the late closure age anomalies. We propose to rewrite the sentence to: "As explained Section 4.1 late pore closure tends to produce weaker age anomalies than early closure, due to the sealing of the surrounding firn. To take into account the lack of explicit gas transport in the model, we reduce the standard deviation of late closure age anomalies to 52 yrs, i.e. 25% of the value used for early closure artifacts."

*-p15 l13:* Comment on the term 'chronological'

By chronological signal we mean the signal chronologically monotonous and representative of atmospheric variability only. We propose to change the term chronological signal to "chronologically monotonous signal, free of layering artifacts".

*- p17 l8: a log normal distribution is not a true representation of the GAD, only a (not so good but simple) approximation. As long as you compare log-normal distribution of different GADs with each other, this seems o.k., but to my understanding the Witrant age distribution is per se not log-normal so a comparison may not be fair. You should mention that explicitly in the text!!!*

Indeed, Witrant et al. do not propose log-normal formulation of their Vostok GAD. However to facilitate comparison with other GADs we fit the Witrant et al. GAD by a log-normal distribution.

The smoothing of the true Witrant GAD is virtually identical to the log-normal fit. This is displayed in the figure below (CFA measurements in blue, smoothing of WDC by our optimal GAD in black, by the true Witrant et al. GAD in brown, and by the log-normal fit to Witrant et al. in yellow). We added this fact in the new manuscript p16 l11.

[Figure]

*- p18 l27: However the outcome of your smoothing is depending on your assumption of a log-normal distribution. You may end up with a simlar record using a different filter shape. I don't think this is critical here but should be kept in mind.*

Yes we are dependent of the log-normal assumption. A log-normal filter still manages to produce a record similar to the observed one within uncertainties for the DO-17 at Vostok. However in the analysis of future signals a log-normal distribution might be insufficient.

*- Figure 6: The true GAD would be the multiplication of the relative Vostok GAD and the Absolute WDC GAD as your WDC signal is already slightly smoothed. As the WDC GAD is very narrow, this does not invalidate your conclusions but should be mentioned!*

Yes mathematically speaking the Vostok GAD is the convolution product between the WDC GAD and our log-normal distribution. Since the WDC GAD is narrow, it does not significantly affect our result. This is another way to formulate the difference in cutting frequencies described p18 l18.
We modified the sentence p15l22 of the new manuscript to reflect this:

[revised manuscript text omitted]